# Influenza A virus induces PI4P production at the endoplasmic reticulum in an ATG16L1-dependent manner to promote the egress of viral ribonucleoproteins

Carla Alemany[1‡], Juliane Da Graça[1‡], Quentin Giai Gianetto[2,3], Maud Dupont[4], Sylvain Paisant[4], Thibaut Douché[2], Catherine Isel[4], Cédric Delevoye[1], Lydia Danglot[5,6], Mariette Matondo[2], Etienne Morel[1‡*], Jean-Baptiste Brault[4‡¤*], Nadia Naffakh[4‡*]

**1** INSERM U1151, CNRS UMR8253, Institut Necker Enfants Malades, Université Paris Cité, Paris, France, **2** Institut Pasteur, Université Paris Cité, CNRS UAR2024, Mass Spectrometry for Biology Unit, Proteomic Platform, Paris, France, **3** Institut Pasteur, Université Paris Cité, Bioinformatics and Biostatistics Hub, Paris, France, **4** Institut Pasteur, Université Paris Cité, CNRS UMR3569, RNA Biology and Influenza Virus, Paris, France, **5** Université Paris Cité, Institute of Psychiatry and Neuroscience of Paris (IPNP), INSERM U1266, NeurImag Core Facility, Paris, France, **6** Université Paris Cité, Institute of Psychiatry and Neuroscience of Paris (IPNP), INSERM U1266, Membrane Traffic and Diseased Brain, Paris, France

‡ CA and JDG share first authorship on this work. EM, JBB, and NN share last authorship on this work.
¤ Current address: Sorbonne Université, Institut Curie, PSL University, CNRS UMR144, Cell Biology and Cancer, Paris, France
* etienne.morel@inserm.fr (EM); jean-baptiste.brault@curie.fr (JBB); nadia.naffakh@pasteur.fr (NN)

## Abstract

The genomic RNAs of influenza A viruses (IAVs) are replicated in the nucleus of infected cells in the form of viral ribonucleoproteins (vRNPs) before being exported to the cytoplasm. The small GTPase RAB11A is involved in the transport of vRNPs to the sites of viral assembly at the plasma membrane, but the molecular mechanisms involved remain largely unknown. Here we show that IAV infection remodels the architecture of the endoplasmic reticulum (ER) sheets, where vRNPs tend to accumulate in the absence of RAB11A. To decipher the interplay between RAB11A, vRNPs, and the ER, we investigated viral-induced perturbations of RAB11A proximity interactome. To this end, we generated cells stably expressing a TurboID-RAB11A fusion protein and performed biotin-based proximity labeling upon viral infection. We found that cellular regulators of phophatidylinositol-4-phosphate (PI4P) homeostasis, including the autophagic and stress response protein ATG16L1, are significantly enriched at the vicinity of RAB11A in infected cells. Infection induces an increase in cellular PI4P levels in an ATG16L1-dependent manner, while ATG16L1 relocalizes to ER membranes upon infection. Depletion of ATG16L1 decreases the co-distribution of vRNPs with PI4P punctae on ER membranes, and reduces the accumulation of vRNPs at the plasma membrane as well as the production of IAV infectious particles. Our data extend to IAVs the notion that viruses can modulate the metabolism and localization of phosphoinositides to control host membrane dynamics and point

**Data availability statement:** The mass-spectrometry proteomics data are available at the PRIDE repository with the dataset identifier PXD053858. The raw microscopy images are available at the Zenodo repository (https://zenodo.org/records/15682874) All other raw data are within the manuscript and its Supporting information files : S1–S3 Files (numerical data for mass spectrometry data analyses), S4 File (uncropped western blots), S5 File (primer sequences) and S6 File (numerical data for graphs).

**Funding:** This work was funded by the Agence Nationale de la Recherche (ANR-21-CE11-0010-03 to NN and LD, ANR-10-LABX-62 to MM and NN, ANR-19-CE16-0012 to LD, ANR-17-CE140030-02, ANR-17-CE13-0015-003, ANR 22-CE14-0019 and ANR 18-CE14-0006 to EM, and ANR-21-CE35-0007 to MM), the Human Frontiers Science Program (HFSP RPG0040/2019 to NN), the Fondation pour la Recherche Médicale (FRM, « labellisation équipe » to EM), and the DIM 1Health to MM. JBB was supported by the HFSP RPG0040/2019 and the ANR-21-CE11-0010-03 grants. CA was supported by the ANR 22-CE14-0019 grant. JDG is a recipient of a doctoral fellowship from the French Ministry of Research/Université Paris-Cité and a 4th year PhD FRM scholarship (grant FDT202304016558). The funders did not play any role in the study design, data collection and analysis, decision to publish, or preparation of the manuscript.

**Competing interests:** The authors have declared that no competing interests exist.

**Abbreviations:** ACN, anhydrous acetonitrile; AF, Alexa Fluor; CHX, cycloheximide; CLIMP63, cytoskeleton-linking membrane protein 63; DMEM, Dulbecco's modified Eagle's medium; ER, endoplasmic reticulum; FA, formic acid; FDR, false discovery rate; GO, Gene Ontology; HA, hemagglutinin; IAVs, influenza A viruses; ICVs, irregularly coated vesicles; MDCK, Madin-Darby Canine Kidney; MEM, Modified Eagle's Medium; MTOC, microtubule organizing center; NP, nucleoprotein; NT, Non-Target; PI, phosphoinositide; PIP$_3$, phosphatidylinositol triphosphate; PI4K, phosphaditylinositol-4 kinases; PI4P, phophatidylinositol-4-phosphate; RTN3, reticulon 3 protein; STED, Stimulated Emission Depletion Microscopy; vRNAs, viral RNAs; vRNPs, viral ribonucleoproteins; ZIKV, Zika virus.

to the ER as an essential platform for vRNP transport. They provide evidence for a pivotal role of ATG16L1 in regulating the identity of endomembranes and coordinating RAB11A and PI4P-enriched membranes to ensure delivery of vRNPs to the plasma membrane.

## Introduction

Influenza A viruses (IAVs) present continuous animal and public health challenges. Human-adapted IAVs recur every year due to antigenic variation, as seasonal IAVs [1]. Wild aquatic birds and domestic species are hosts to a dynamic pool of IAVs, which are responsible for epizootic, zoonotic, and potentially pandemic outbreaks [2]. Sporadically, as a consequence of the segmentation of their genome into a bundle made of eight distinct viral RNAs (vRNAs), novel and possibly devastating IAVs are generated through co-infection and genetic mixing of vRNAs from animal and human IAVs. The molecular mechanisms of vRNA intracellular transport from the nucleus to the plasma membrane and vRNA assembly into bundles, which are critical for reassortment, are only partially understood [3]. Elucidating this aspect of the viral cycle will help in the broader goal to achieve better prevention and treatment of the disease.

Each influenza vRNA, a single-stranded RNA of negative polarity, is associated with viral proteins into macromolecular complexes called viral ribonucleoproteins (vRNPs) that are 30–120 nm in length and 12–15 nm in diameter. Within each vRNP, the vRNA adopts a closed conformation by base-pairing of the 3′ and 5′ ends. The resulting ~15 bp-long duplex associates with one copy of the viral polymerase, a PB1-PB2-PA heterotrimer, while the remaining RNA is bound by the nucleoprotein (NP). Upon viral entry, vRNPs are released in the cytoplasm and transported into the nucleus, where they serve as templates for transcription and replication of the viral genome [4]. Neo-synthesized vRNPs may become templates for new rounds of transcription/replication, or exit the nucleus through the CRM1-dependent nuclear export pathway. They are then transported toward the plasma membrane where they get assembled with other viral proteins and glycoproteins, enabling the budding of new virions [3].

There is much evidence that the RAB11A small GTPase is involved in vRNP trafficking and assembly of IAV genome, including the fact that it interacts directly with the PB2 component of vRNPs (reviewed in [5,6]). After exiting the nucleus, vRNPs accumulate in a perinuclear region close to the microtubule organizing center (MTOC), where they co-localize with RAB11A-positive membranes. At later time points in infection, inclusions containing vRNPs and RAB11A can be observed throughout the cytoplasm [7]. RAB11A is a key player of (and used as a marker for) recycling endosomes and its best-documented function is to regulate the recycling of internalized proteins, from endosomes to the plasma membrane [8]. Therefore, it was initially proposed that RAB11A mediates the docking of vRNPs to recycling endosomes in the vicinity of the MTOC, after which recycling endosomes carry vRNPs

toward the plasma membrane along microtubules. However, this model is now called into question by the fact that infected cells show (i) alterations in the efficiency of the RAB11A-mediated recycling pathway, very likely due to the fact vRNPs hinder RAB11A binding to RAB11-FIP effectors [9,10], and (ii) alterations in the intracellular distribution of RAB11A, which is found in close proximity with endoplasmic reticulum (ER) membranes [11,12]. It was proposed that the disruption of RAB11A canonical function leads to the concentration of recycling endosomes coated with vRNPs in condensates as a mechanism to facilitate vRNP bundling and assembly of the viral genome [10,11].

Alongside recycling endosomes, the ER compartment is also altered in IAV-infected cells. We previously reported that IAV infection induces a progressive remodeling and tubulation of ER membranes around the MTOC and all throughout the cell [12]. RAB11A and vRNPs were detected close to the ER membranes and at the surface of a new type of RAB11A-positive vesicles, distinct of recycling endosomes, which were named irregularly coated vesicles (ICVs). Some ICVs are observed very close to the ER or the plasma membrane, suggesting that these vesicles could be transporting vRNPs between the two membrane compartments. It was recently shown that the viral hemagglutinin (HA), a glycoprotein which travels through the ER-Golgi secretory pathway, drives the formation of remodeled membrane compartments on which vRNPs tend to accumulate [13]. Interestingly, expression of a dominant-negative mutant of RAB11A decreases vRNP association with HA-remodeled membranes [13] as well as the formation of ICVs [12].

Many questions remain to be answered, notably regarding the site(s) at which RAB11A and vRNPs cooperate, the biogenesis of vRNP-coated vesicles, and their subsequent transport to virion assembly sites at the plasma membrane. Here, we aimed at unraveling the interplay between RAB11A, the ER membranes, and vRNPs. Using a combination of confocal and STED microscopy, we extend our previous observations and show that IAV infection specifically alters the distribution of ER sheets. We find that in RAB11A-depleted cells, vRNPs tend to accumulate at the vicinity of remodeled ER membranes. To decipher the interplay between RAB11A, vRNPs, and the ER, we combined proximity-labeling and affinity purification-mass spectrometry on cells stably expressing a TurboID-RAB11A fusion protein. We found that several cellular regulators of phosphoinositide (PI) homeostasis, including the autophagy-related protein ATG16L1 [14], are enriched in the proximity-interactome of RAB11A in IAV-infected compared to uninfected cells.

PIs represent a minor fraction of total cellular phospholipids, yet they play key roles in mediating membrane signal transduction, shaping and defining the membrane identity of organelles, and regulating vesicular trafficking (reviewed in [15,16]). PIs are produced from a phosphatidylinositol precursor, the inositol ring of which can be phosphorylated at the D3, D4, and D5 positions. As a result, distinct phosphadityilinositol monophosphates (PI3P, PI4P, and PI5P), phosphadityilinositol biphosphates (PI(3,4)$P_2$, PI(3,5)$P_2$, PI(4,5)$P_2$) and one phosphadityilinositol triphosphate (PIP$_3$) can be produced. Multiple kinases, phosphatases, and regulatory proteins are controlling the spatial and temporal turnover of PIs, and the specific composition and dynamics of PI pools at specific endomembrane sites. The cross-talk between PIs and RAB GTPases plays a major role in maintaining this "membrane code" and controlling membrane remodeling and trafficking [17].

Several positive-sense RNA viruses manipulate the PI regulatory machinery and hijack phosphadityilinositol-4 kinases (PI4K) to induce the formation of modified, PI4P-enriched endomembrane compartments referred to as viral replication organelles (reviewed in [18]). However, it remained unknown whether PI metabolism is rewired during IAV infection and whether it plays a role in vRNP trafficking. Here, we show that the PI3P/PI4P balance is altered and the pool of PI4P associated with the ER increases upon IAV-infection, through a pathway that involves ATG16L1. Depletion of ATG16L1 affects the co-distribution of PI4P punctae with vRNPs on ER membranes, delays the accumulation of vRNPs at the plasma membrane, and decreases the production of IAV infectious particles. Hence, we propose a working model in which the vRNPs get associated to a remodeled ER where ATG16L1 mediates a local production of PI4P, thereby regulating the membrane identity of vRNP-coated vesicles and, together with RAB11A, promoting their transport to the plasma membrane.

## Results

### IAV infection specifically remodels the ER sheets architecture

We sought to extend previous evidence that the ER compartment is altered in IAV-infected cells. To this end, we performed confocal microscopy using antibodies specific for the cytoskeleton-linking membrane protein 63 (CLIMP63) and the reticulon 3 protein (RTN3), known to be enriched in ER sheets and ER tubules, respectively [19,20]. A549 cells were infected at a high multiplicity of infection with the A/WSN/33 (WSN) or A/Victoria/3/75 (VIC) virus, fixed at 8 h post-infection (hpi) and stained for the endogenous RTN3 and CLIMP63 proteins, and the viral NP. The NP staining is both nuclear and cytoplasmic, as expected during the late phase of the viral life cycle. The RTN3 staining shows no significant alteration in IAV-infected compared to mock-infected cells (Fig 1A). In contrast, the CLIMP63 staining is markedly altered upon IAV infection, exhibiting an irregular and very extended distribution compared to the dense and homogeneous perinuclear signal in mock-infected cells.

We quantified the IAV-induced remodeling of ER sheets using the cell image analysis software Cell Profiler [21,22]. Cells were segmented into an RTN3+ area, used as a proxy for the total cell area (delineated by a dotted line in Fig 1A), and another CLIMP63+ area. The ratio of the CLIMP63+ area over the RTN3+ area is significantly higher in WSN- and VIC-infected cells (median values of 0.65±0.20 and 0.72±0.16, respectively) than in mock-infected cells (0.46±0.17, $p < 0.001$) (Fig 1B). In the same conditions of infection, the steady-state levels of CLIMP63, as assessed by western blot on total cell lysates, remain unchanged (Fig 1C–1D). Therefore, the increase in the CLIMP63+ to RTN3+ area ratio is indicative of a redistribution rather than an over-production or over-accumulation of CLIMP63. The staining for CLIMP63 and the viral HA surface glycoprotein, which travels through the ER-Golgi pathway, largely overlap, indicating that CLIMP63 remains located within the ER (S1A and S1B Fig). The ER remodeling induced by IAV is not merely the consequence of a general stress response induced by viral infections, as cells infected with Zika virus (ZIKV) show a very distinct pattern where both RTN3 and CLIMP63 are relocalized and concentrated in the same area as the NS3 viral antigen, a marker for viral factories (S1C Fig). In summary, our data demonstrate that IAV induces a specific alteration of ER sheet architecture, characterized by an extension toward the cell periphery.

### vRNPs accumulate at the vicinity of ER membranes upon RAB11A depletion

RAB11A depletion leads to an alteration of the late stages of IAV infection, notably the distribution pattern of vRNPs in the cytoplasm, a delayed delivery at the plasma membrane, and a reduction in the production of infectious viral particles [7,23–25]. This points to a role for RAB11A in vRNP transport, but its exact mechanism of action and how it is related to ER alterations remain unclear. Here, we assessed the impact of RAB11A depletion on the remodeling of CLIMP63+ ER membranes and their proximity with vRNPs. As the NP staining largely coincides with PB2 protein (S2A and S2B Fig), NP signal was used as a proxy to visualize vRNPs in the cytoplasm. Prior to infection, A549 cells were treated with RAB11A-specific siRNAs or Non-Target (NT) siRNAs, and RAB11A knock-down efficiency was confirmed by western blot analysis (S6C Fig) and by immunofluorescence (Figs 2A and S6A). The depletion of RAB11A prevents the formation of large cytoplasmic inclusions of vRNPs upon viral infection (Fig 2A), as expected from previous studies [7,23], but does not affect the remodeling of CLIMP63+ membranes (Figs 2A and S2C). Strikingly, in RAB11A-depleted cells the viral NP signal is found at the vicinity of CLIMP63+ membranes in the perinuclear region (Fig 2A and 2B). While a similar distribution of NP in a reticulated pattern in the perinuclear region of RAB11A-depleted cells has already been reported by Eisfeld and colleagues [7], our observations suggest that the sites where vRNPs accumulate in the absence of RAB11A correspond to ER membranes.

To further document this point, we used Stimulated Emission Depletion Microscopy (STED), which overcomes the diffraction-limited resolution of confocal microscopes and enables super-resolution imaging [26]. To this end, we used a U20S cell line characterized by a large and flat cytoplasmic area, stably expressing the ER protein Sec61ß fused to

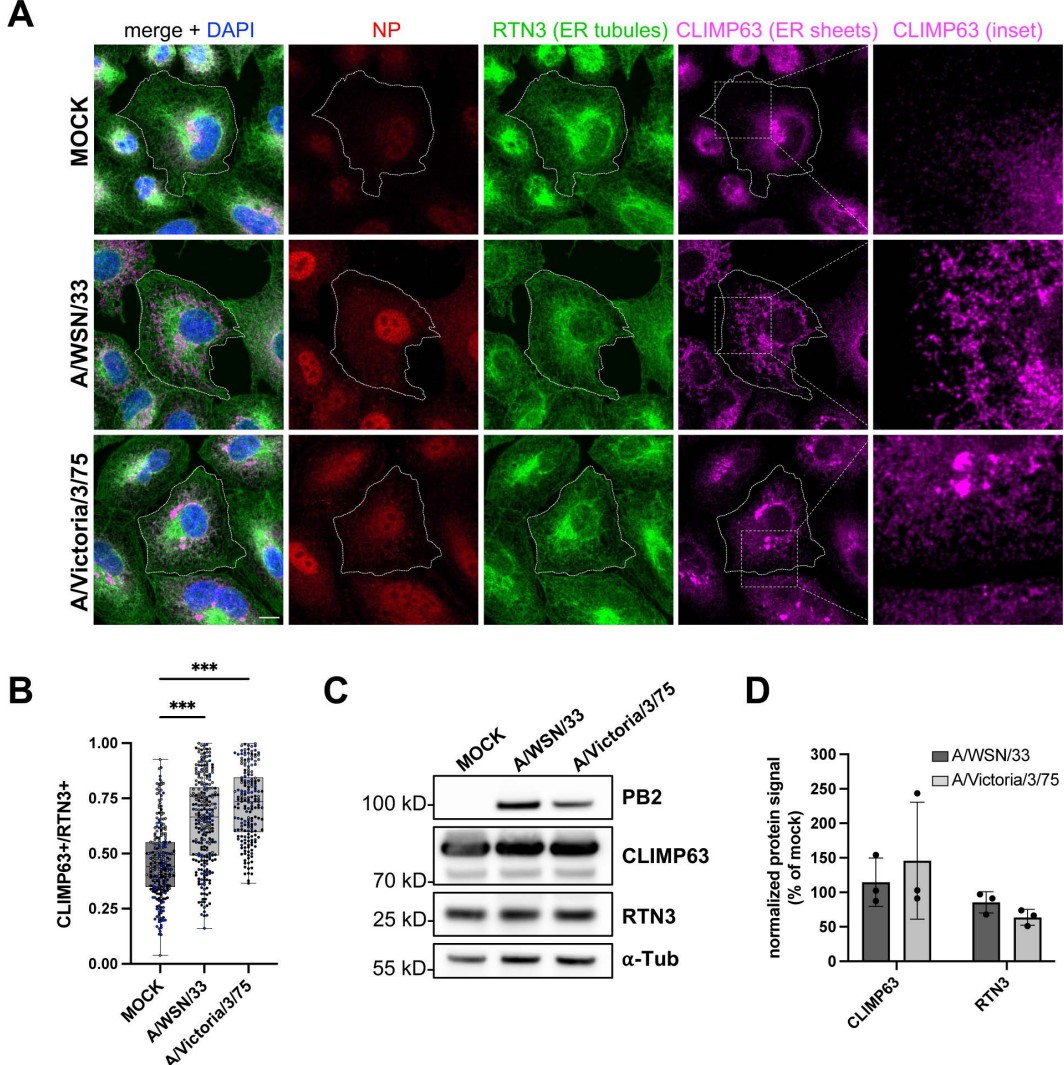

**Fig 1. IAV infection specifically remodels ER sheets. A.** A549 cells were infected with WSN or VIC at a MOI of 5 PFU/cell for 8 h, or mock-infected. Fixed cells were stained for the viral NP and the cellular markers for ER sheets and ER tubules, CLIMP63 and RTN3, respectively. The RTN3 staining was used to delineate the cell edges. Nuclei were stained with DAPI (blue), and cells were imaged with a confocal microscope. Scale bar: 10 µm. **B.** A549 cells treated as in (A) were analyzed with the Cell Profiler software in order to segment CLIMP63+ and RTN3+ areas based on the Otsu thresholding method. The ratios of the CLIMP63+ area to the RTN3+ area are shown. Each dot represents one cell, and the data from three independent experiments are shown (black, gray and blue dots). The median and interquartile values are represented as box-plots (182-262 cells per condition). ***: $p$-value < 0.001, one-way ANOVA. **C, D.** A549 cells were infected or mock-infected as in (A). At 8 hpi, total cell lysates were prepared and analyzed by western blot, using the indicated antibodies. (C) Cropped blots of one representative experiment out of three are shown. (D) The signals for CLIMP63 and RTN3 are normalized over the α-tubulin signal and expressed as percentages (100%: mock-infected cells). The data shown are the mean ± SD of three independent experiments. No significant difference is detected between infected and mock-infected cells (two-way ANOVA with Sidak's multiple comparison test). The data underlying this figure can be found at https://zenodo.org/records/15682874 (raw images), in S4 File (uncropped western blots) and S6 File (graphs raw data).

mEmerald, and therefore particularly convenient for imaging of the ER [27]. Overexpressed Sec61ß was shown to label both the ER sheets and tubules [19]. This method allowed us to confirm that in RAB11A-depleted cells, vRNPs can be observed along membrane structures in the immediate vicinity of Sec61ß-positive membranes (Fig 2C). The Pearson correlation coefficient was used to assess the covariance of Sec61ß and NP signal levels in control and RAB11A-depleted

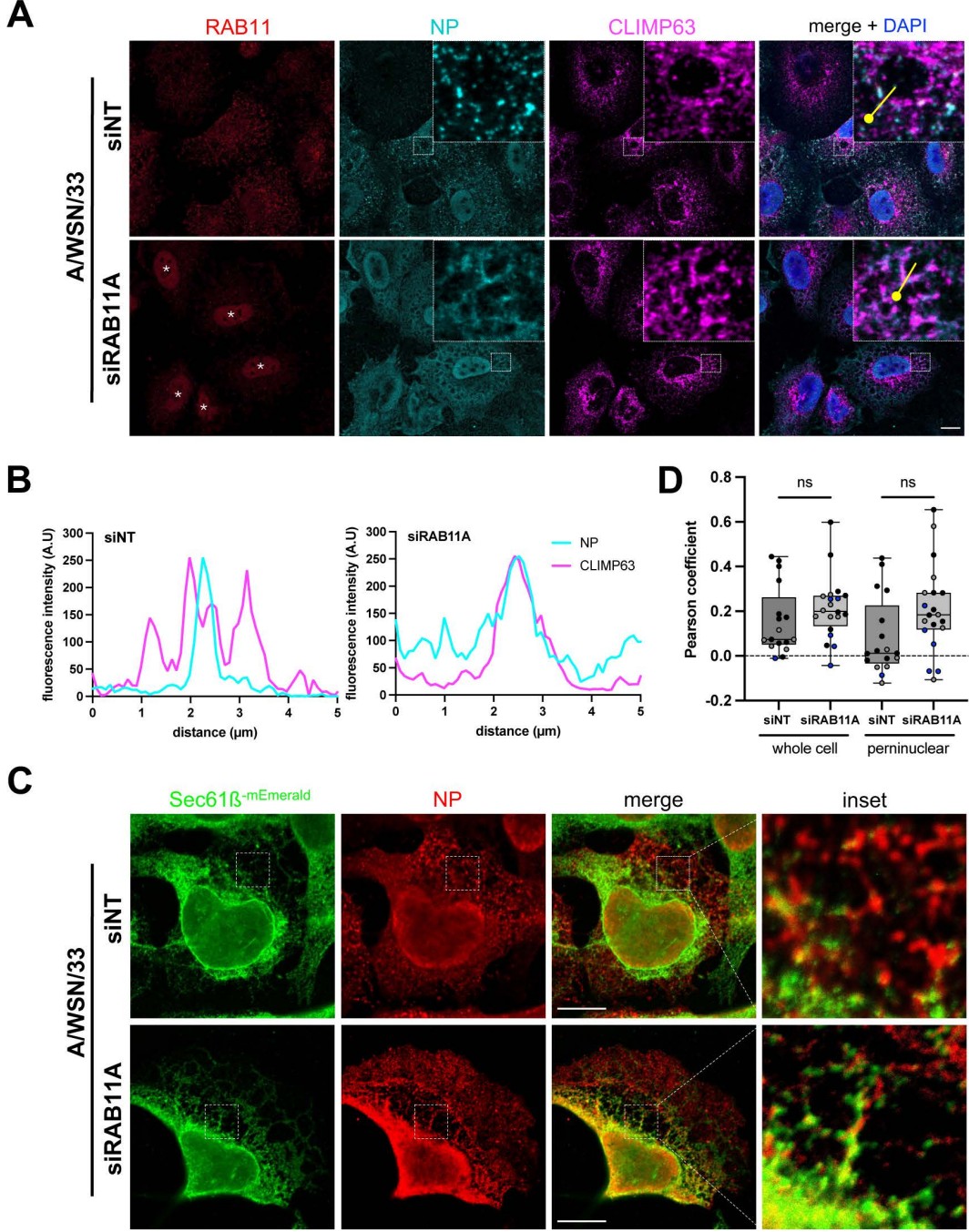

**Fig 2. Impact of RAB11A depletion on ER remodeling and vRNP localization upon infection with the WSN virus. A.** A549 cells were treated with RAB11A-specific or control Non-Target (NT) siRNAs for 48 h, and subsequently infected with WSN at a MOI of 5 PFU/cell for 4 h. Fixed cells were stained for the viral NP and the cellular RAB11 and CLIMP63 proteins. Nuclei were stained with DAPI (blue), and cells were imaged with a confocal microscope. White stars: non-specific nuclear staining visible upon siRNA treatment, unrelated to NP antibody bleed-through as documented in S6A Fig. Scale bar: 10 μm. **B.** Fluorescence intensity profiles for NP (cyan) and CLIMP63 (magenta) along the yellow lines drawn in panel (A) (merge insets), starting from the knob. **C.** U2OS cells stably expressing the ER translocon Sec61ß fused to mEmerald (U2OS-Sec61ß-mEmerald) were treated with RAB11A-specific or control Non-Target (NT) siRNAs for 48 h, and subsequently infected with WSN at a MOI of 5 PFU/cell for 8 h. Fixed cells were stained for NP and Sec61ß-mEmerald (anti-GFP antibody) and images were acquired using STED microscopy. Scale bar: 10 μm. **D.** U2OS-Sec61ß-mEmerald cells treated as in (A) were analyzed to assess co-localization of NP and Sec61ß, using a pixel-based method to determine the Pearson

cells. We computed the Pearson coefficient without any thresholding, therefore taking into consideration both the diffuse and aggregated NP signal. The data showed low Pearson coefficient values, which is consistent with the fact that only a fraction of the NP signal is observed at the vicinity of ER membranes, with the NP and Sec61ß signals showing partial overlap. Interestingly however, we observed a non-significant trend to increase in RAB11A-depleted cells (median values of 0.07 and 0.20 in siNT- and siRAB11A-treated cells, respectively), this trend being more pronounced in the perinuclear region (0.01 and 0.18 in siNT- and siRAB11A-treated cells, respectively) (Fig 2D).

Taken together, our data strongly suggest that after exiting the nucleus, vRNPs are targeted to ER membranes independently of RAB11A.

### The proximity interactome of RAB11A is enriched in ATG16L1 and phosphoinositide-metabolizing enzymes upon IAV infection

To decipher the interplay between vRNPs, the ER and RAB11A, we determined to what extent its protein interactome is modified upon IAV infection. To this end, we used a TurboID-based proximity labeling approach, most suitable for the detection of weak or transient protein-protein interactions (Fig 3A). TurboID is a highly active biotin ligase, it rapidly converts biotin into biotin–AMP, a reactive intermediate which covalently labels neighboring proteins within a 10–30 nm labeling radius [28]. The RAB11A protein fused to TurboID was stably expressed in A549 cells by lentiviral transduction, and we isolated a clonal population of A549-TurboID-RAB11A cells showing comparable steady-state levels for the recombinant TurboID-RAB11A and the endogenous RAB11A proteins, as assessed by western blot (S3A Fig). Upon immunostaining, the TurboID-RAB11A protein showed the same subcellular distribution as the endogenous RAB11A, *i.e.,* regular cytoplasmic punctae concentrated in a perinuclear area in uninfected cells, larger and irregular cytoplasmic punctae strongly co-localizing with the viral NP in infected cells (S3B Fig). Addition of biotin to the culture medium during 10 min led to a detectable accumulation of biotinylated proteins (S3C Fig).

The A549-TurboID-RAB11A clonal cells were infected with the WSN virus at a MOI of 5 PFU/cell or mock-infected. At 9 hpi, biotin was added to the culture medium, cell lysates were prepared and biotinylated proteins were purified via streptavidin beads, as described in [29]. A fraction of the total lysates and the biotinylated eluates were then analyzed by mass spectrometry-based proteomics.

Correlation analysis of the mass spectrometry data showed that the independent biological replicates were highly consistent, with Pearson correlation coefficients between the protein intensity values higher than 86.76% and 96.02% for the WSN- and mock-infected samples (*n* = 4), respectively (S3D Fig). Moreover, the distribution of protein intensity values across the different samples was very similar (S3E Fig). Gene Ontology (GO) enrichment analysis revealed a significant enrichment in GO terms related to membrane-associated factors, as well as cellular localization, transport, and vesicles within the 3775 proteins found to be enriched in biotinylated eluates (S4A Fig and S1 File), which provides evidence for the specificity of our proximity labeling dataset.

Upon differential analysis of the biotinylated eluates, using a fold-change threshold of 2 and a false discovery rate (FDR) threshold of 1%, we found that 2,382 and 407 cellular proteins were less abundant and more abundant, respectively, in the eluates derived from WSN-infected cells compared to the mock-infected controls (Fig 3A, green and blue color, respectively, and S2 File). Importantly, 1901 out of 2,382 and 370 out of 407 proteins did not show the same trend of lower or greater abundance, respectively, when total lysates of WSN- and mock-infected cells were compared (S4B Fig and S3 File). Therefore, the variations in protein abundance observed in biotinylated eluates, whether they derive from

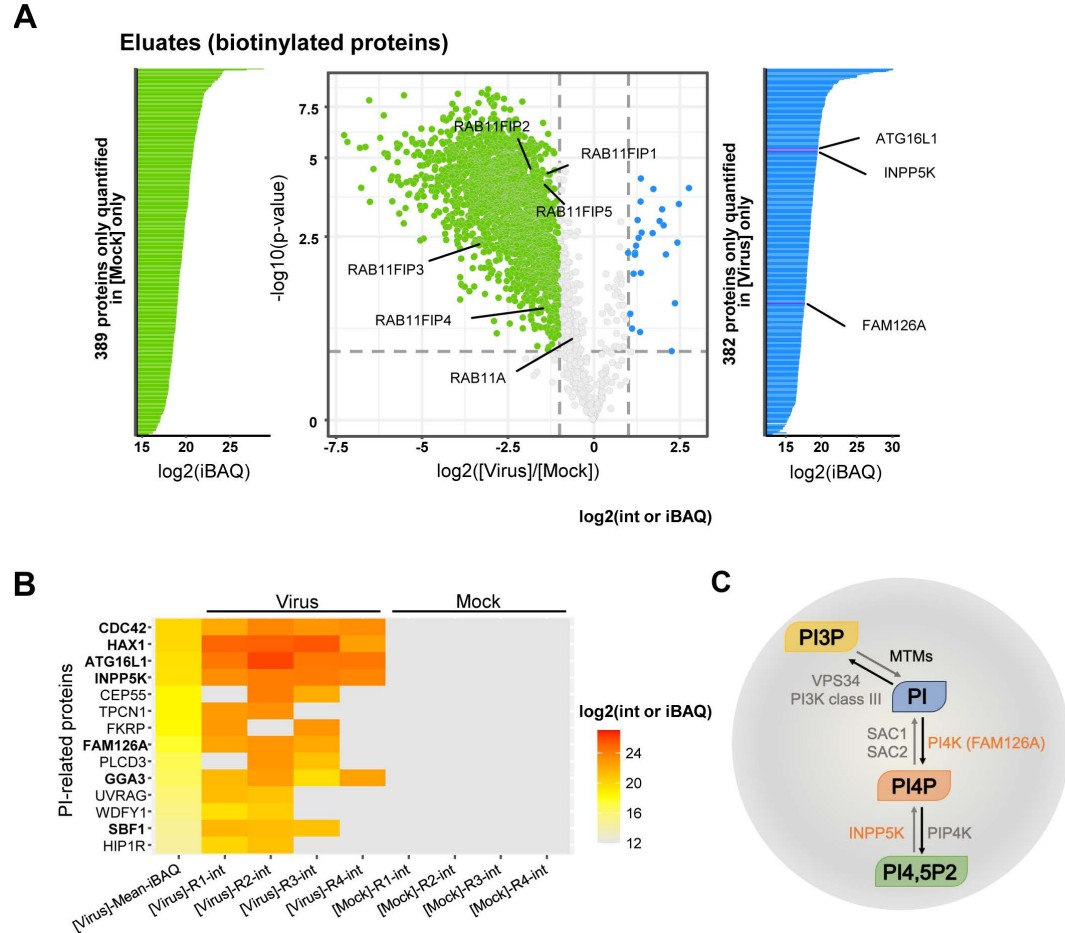

**Fig 3. Viral-induced perturbations of RAB11A proximity interactome. A.** Volcano plot showing the log2 fold change (*x* axis) and its significance (−log10(*p*-value), *y* axis) associated to a false discovery rate <1%, for each protein (dots) in eluates from the RAB11A proximity labeling experiment. The log2 fold change refers to the enrichment in WSN-infected (*n*=4) vs. mock-infected (*n*=4) samples. Blue and green dots represent proteins enriched in WSN-infected vs. mock-infected samples, and proteins enriched in mock-infected vs. WSN-infected samples, respectively. The iBAQ (intensity Based Absolute Quantification) plots shown on the sides of the volcano plot provide additional information on proteins for which no statistical comparison of the abundance could be performed (hence they are not represented in the volcano plot), because they are present only in WSN-infected samples (blue) or in mock-infected samples only (green). **B.** Heat-map of cellular proteins associated with a GO term or a description containing the term "phosphatidylinositol", according to the intensity values measured in the IAV-infected (*n*=4) or mock-infected (*n*=4) samples. On the left side of the heatmap, the "mean iBAQ in IAV" column represents the mean of the iBAQ values in the IAV-infected samples. The iBAQ value approximates the abundance of a protein by dividing the (total precursor) intensities by the number of theoretically observable tryptic peptides of the protein. The same log2 scale is used for intensity values of the replicates and for mean iBAQ values (shown on the right). **C.** Schematic representation of the enzymatic regulation of the PI3P/PI4P balance. Black and gray arrows represent the activities of the indicated kinases and phosphatase-related proteins, respectively. MTMs: myotubularins; INPP5K: inositol polyphosphate-5-phosphatase K; PI3K, PI4K, PI4PK: PI3-, PI4- and PI4P kinases, respectively; SAC1 and SAC2: PI4P phosphatases. The data underlying this figure can be found in S2 File.

infected or uninfected cells, most likely reflect specific, viral-induced changes. The large number of proteins with reduced or increased abundance demonstrate that the proximity interactome of RAB11A is strongly disrupted upon infection. Strikingly, all members of the RAB11-family interacting proteins, known to interact with RAB11 to mediate the endosomal recycling pathway [30], i.e., RAB11FIP-1, -2, -3, -4, and -5, show reduced abundance (Fig 3A). This observation is fully in line with previous reports showing that the endosomal recycling function of RAB11 is impaired in IAV-infected cells [9,10].

Among the 370 proteins with increased abundance in RAB11A proximity interactome, we found a significant enrichment of GO terms related to mitochondrial proteins, notably mitochondrial ribosomal proteins (S2 File). This enrichment could possibly relate to a previous report that IAV infection alters the morphodynamics of mitochondria [31]. Interestingly, several proteins known to regulate the PIs metabolism were exclusively quantified in biotinylated eluates derived from IAV-infected cells but not in those derived from mock-infected cells (Fig 3B). Seven proteins (CDC42, HAX1, ATG16L1, INPP5K, FAM126A, GGA3, and SBF1) were notably identified in at least three out of the four replicates of infected biotinylated eluates, *i.e.,* with a good reproducibility. Among these, two proteins are direct enzymatic regulators of the synthesis of the mono-phosphate PI4P (Fig 3C), i.e. FAM126A (also known as HYCC1, a subunit of the phosphatidylinositol 4-kinase complex PI4KIIIA [32]) and INPP5K (a 5-P phosphatase producing mostly PI4P [33]), while ATG16L1, an autophagy-related protein, was shown to modulate the production and trafficking of PI4P [34]. These findings suggest a potential link between PI4P homeostasis and RAB11A-mediated transport of vRNPs.

## IAV infection induces an increase in PI4P production at the ER

Our finding that upon IAV infection, RAB11A environment becomes enriched in proteins associated with PIs metabolism is consistent with the fact that PIs, especially mono-phosphates such as PI3P and PI4P, were reported to be closely associated with RAB11A-positive membranes [35] and to play an essential role in stress-dependent regulation of endomembrane morphodynamics [16,35,36]. These observations prompted us to investigate a possible alteration of the PIs balance upon IAV infection at a high MOI, i.e., in single-cycle conditions. We analyzed the global levels of PI3P and PI4P using fluorescence-based methods, and compared the mean PI3P or PI4P signal intensity per cell in WSN-infected compared to mock-infected A549 cells. At 8 hpi, using a FYVE-GST purified peptide [37] to label PI3P-positive membranes, we detected a sharp decrease of the PI3P signal in infected cells compared to control cells ($p < 0.0001$) (Fig 4A and 4B, left panels). In contrast, using either the natural PI4P binder SidC fused to the SNAP-tag as a biosensor [38] or an anti-PI4P antibody, we detected a stronger PI4P signal in the cytoplasm of infected cells compared to control cells ($p < 0.0001$) (Fig 4A and 4B, right panels and S5A and S5B Fig, respectively). This trend was already very pronounced at 4 hpi (S7F and S7G Fig). Noticeably, the Golgi pool of PI4P, which is detected with the biosensor but not with the antibody, does not seem to be affected upon viral infection. To investigate the subcellular distribution of this increased PI4P signal, we monitored its putative association with ER membranes using U2OS-mEmerald-Sec61ß cells. Whether detected with the PI4P-specific biosensor or antibody (Fig 4C, 4D and S5C, S5D, respectively), the few PI4P-positive punctae in mock-infected cells are dispersed throughout the cytoplasm, whereas PI4P cytoplasmic punctae formed upon IAV infection at 8 hpi are mostly associated with the ER.

We then performed siRNA-mediated knockdown of the two RAB11A proximity-labeling hits known to enzymatically regulate the PIs metabolism in favor of PI4P synthesis, FAM126A and INPP5K (S6B Fig), and analyzed the global levels of PI4P upon IAV infection. Interestingly, the depletion of one or the other of these proteins abolishes the PI4P increase that we reported in IAV-infected cells (Figs 4E, 4F and S5E, S5F). Similar results were obtained by knocking down the autophagic ATG16L1 protein (S6C Fig), also found enriched in the vicinity of RAB11A upon infection in our proximity labeling experiments, and known to be involved in PI4P turnover [34] (Figs 4G, 4H and S5G, S5H). None of the siRNAs had any significant effect on cell viability (S6D Fig).

The progression of the viral cycle was monitored in cells depleted of ATG16L1, FAM126A, or INPP5K, in single-cycle conditions. RAB11A-depleted cells were used as a reference. First, strand-specific RTqPCR was used to monitor the accumulation of viral NP transcripts in the presence of cycloheximide (CHX). In these conditions, where no translation of viral proteins can occur, vRNPs are not replicated, and only the parental vRNPs can act as templates for transcription. Therefore, the viral transcripts represent an accurate measure for the efficiency of early stages of the viral cycle, encompassing viral uptake, nuclear import of vRNPs and primary transcription. As indicated by unchanged levels of accumulation of primary NP transcripts in the presence of CHX (S7A and S7B Fig, light gray bars), the earliest stages of the viral

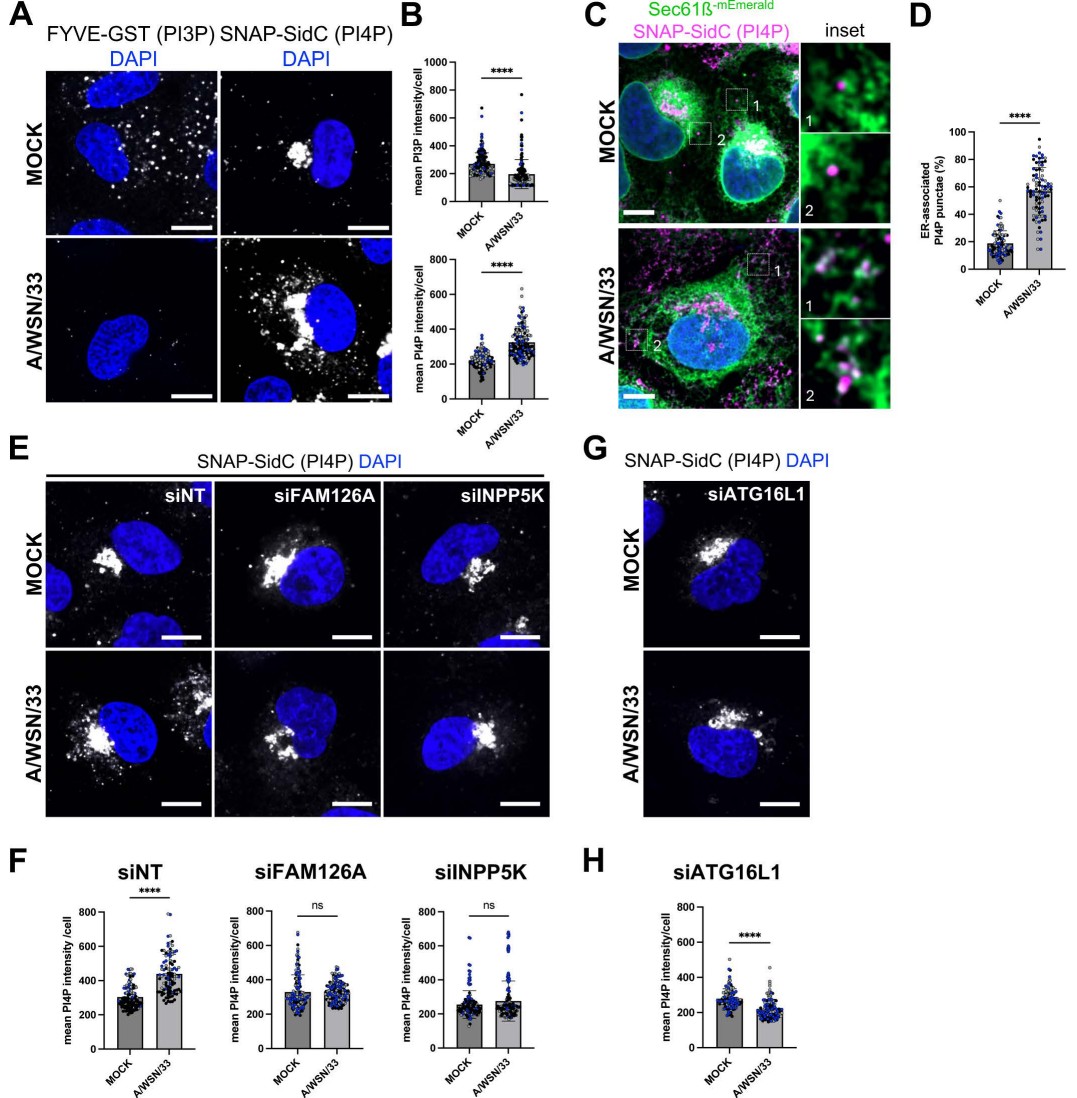

**Fig 4. Viral-induced perturbations of the PI3P/PI4P balance. A.** A549 cells were infected with WSN at a MOI of 5 PFU/cell for 8 h, or mock-infected. Fixed cells were stained for PI3P (FYVE-GST probe) or PI4P (SNAP-SidC probe). Nuclei were stained with DAPI (blue), and cells were imaged with a confocal microscope. Scale bar: 10 μm. **B.** A549 cells treated as in (A) were analyzed with the Fiji software to determine the mean intensity of the PI3P or PI4P signal per cell. Each dot represents one cell, and the data from three independent experiments are shown (black, gray and blue dots). The median and standard deviation values are represented (134-164 cells per condition). ****: *p*-value < 0.0001, unpaired t *test*. **C.** U2OS-Sec61ß-mEmerald cells were infected with WSN at a MOI of 5 PFU/cell for 8 h, or mock-infected. Fixed cells were stained for PI4P as in A and nuclei were stained with DAPI (blue). Cells were imaged with a confocal microscope. Scale bar: 5 μm. **D.** U2OS-Sec61ß-mEmerald cells treated as in (C) were analyzed with the Fiji software to determine the percentage of the total PI4P punctae associated to ER in individual cells. Each dot represents one cell, and the data from three independent experiments are shown (black, gray and blue dots). The mean and standard deviation values are represented as histograms (87–88 cells per condition). ****: *p*-value < 0.0001, unpaired t te*st*. **E, G.** A549 cells were treated with control non-target (NT) siRNAs or with siRNAs targeting FAM126A, INPP5K (E) or ATG16L1 (G) for 48 h, and subsequently infected with WSN at a MOI of 5 PFU/cell for 8 h, or mock-infected. Fixed cells were stained for PI3P or PI4P as in A. Nuclei were stained with DAPI (blue), and cells were imaged with a confocal microscope. Scale bar: 10 μm. **F, H.** A549 cells treated as in (E) and (G), respectively, were analyzed with the Fiji software to determine the mean intensity of the PI3P or PI4P signal per cell. Each dot represents one cell, and the data from three independent experiments are shown (black, gray and blue dots). The median and standard deviation values are represented (98–152 cells per condition). ****: *p*-value < 0.0001, ns: non-significant, unpaired *t* test. The da*ta* underlying this figure can be found at https://zenodo.org/records/15682874 (raw images) and S6 File (graphs raw data).

cycle are not significantly affected by any of the tested siRNA. Then, the accumulation of viral proteins was monitored in the absence of CHX at 5 hpi, using western blot analysis. A moderate (<2-fold) but reproducible decrease in steady-state levels of the HA and NS1 viral proteins was observed in ATG16L1 and INPP5K-depleted cells, but not in FAM126A- or RAB11A-depleted cells (S7C and S7D Fig). In contrast, the depletion of ATG16L1, FAM126A, or INPP5K led to a significant 2- to 6.5-fold reduction in the production of infectious viral particles at 4 and 5 hpi, when virus concentrations >$10^5$ PFU/mL were already measured in the supernatant (S7E Fig).

In summary, our data show that IAV infection induces an increase in PI4P production at the ER, and this increase depends—at least partially—on FAM126A, INPP5K, and ATG16L1. The depletion of any of these three proteins, while having no significant effect at early stages of the viral cycle, reduces the production of infectious viral particles almost to the same extent as the depletion of RAB11A. In the case of RAB11A and FAM126A, the fact that steady-state levels of the viral proteins are unchanged clearly point to a defect at the latest stages (vRNP transport/assembly) of the viral cycle. Although it cannot be formally ruled out that the slight reduction in viral protein levels observed in the absence of ATG16L1 and INPP5K partially accounts for the reduction in viral progeny, we further investigated a possible role for ATG16L1 in vRNP transport and exit.

## ATG16L1 and RAB11A control the abundance, proximity to vRNPs and/or turnover of PI4P pools in IAV-infected cells

As both PI4P (Figs 4C, 4D and S5C, S5D) and vRNPs (Fig 2) are detected at the vicinity of ER membranes in infected cells, we examined the impact of ATG16L1 or RAB11A depletion on the production of PI4P and the distribution of PI4P relative to ER membranes and vRNPs, in IAV-infected U2OS-mEmerald-Sec61ß cells (Fig 5A–5D). The depletion of ATG16L1 led to a significant reduction of the number of PI4P punctae per cell (Fig 5B), in agreement with our previous observations in IAV-infected A549 cells (Figs 4G, 4H and S5G, S5H), with a conserved percentage of ER-associated PI4P punctae (Fig 5C). The depletion of RAB11A also led to a significant reduction of the number of PI4P punctae per cell (Fig 5B), but unlike the depletion of ATG16L1, it resulted in a moderate but significant increase in the percentage of PI4P punctae associated with the ER in infected cells (86 ± 11%, compared to 70 ± 11% in control cells, p < 0.0001) (Fig 5C). These observations suggest that (i) both ATG16L1 and RAB11A contribute to the amplification of PI4P pools at the vicinity of the ER, and (ii) RAB11A contributes to the turnover of these pools, possibly in relation with the biogenesis of RAB11A-positive, vRNP-coated vesicles from PI4P-enriched ER subdomains. To further assess this hypothesis, we measured the percentage of PI4P punctae associated with NP-positive punctae. As shown in Fig 5D, it was 55 ± 14% in control IAV-infected cells, and was reduced to 20 ± 15% in ATG16L1-depleted cells (p < 0.0001). This parameter could not be measured in RAB11A-depleted cells because of the diffuse pattern of the NP signal. Finally, we analyzed the distribution of ATG16L1, and observed that it relocalizes from a predominantly cytoplasmic distribution in mock-infected cells to ER membranes in WSN-infected cells (Fig 5E and 5F). Altogether, our data suggest that ATG16L1 not only contributes to the amplification of PI4P pools at the vicinity of the ER, but also controls their proximity with vRNPs.

Notably, ATG16L1 and RAB11A depletion do not have the same impact on NP distribution (Fig 5A). The punctate pattern of NP in control cells, thought to correspond to clusters of vRNP-coated vesicles [9], becomes diffuse in RAB11A-depleted cells, as previously reported [7,23,25], while it remains unaltered in ATG16L1-depleted cells. The co-localization of vRNPs with RAB11A is also conserved upon ATG16L1 depletion (S8A and S8B Fig). These observations confirm that RAB11A is critical for the biogenesis of vRNP-coated vesicles, and clearly indicate that in the absence of ATG16L1, RAB11A-positive membranes can still support the biogenesis of vRNP-coated vesicles. Therefore, we sought to investigate whether, in the absence of ATG16L1/PI4P signaling, the anterograde transport of vRNP-coated vesicles toward the plasma membrane is impaired.

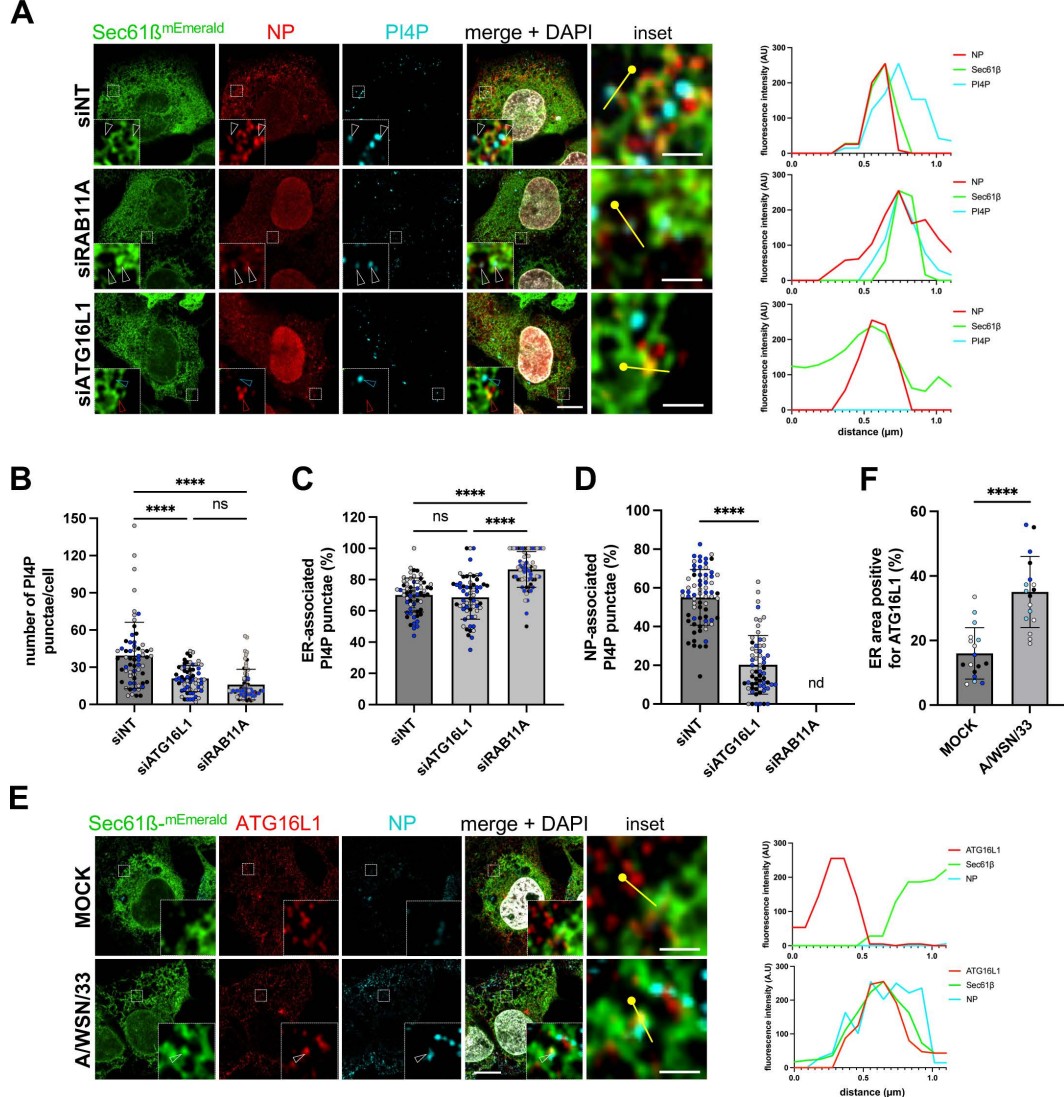

**Fig 5. Impact of RAB11A or ATG16L1 depletion on viral-induced production and localization of PI4P. A.** U2OS-Sec61ß-mEmerald cells were treated with control Non-Target (NT) siRNAs or with siRNAs targeting RAB11A or ATG16L1 for 48 h, and subsequently infected with WSN at a MOI of 5 PFU/cell for 8 h. Fixed cells were immunostained for PI4P and NP, and nuclei were stained with DAPI (white). Cells were imaged with a confocal microscope. Whole cell scale bar: 5 µm; insets scale bar: 1 µm; white arrowheads: NP-Sec61ß-PI4P co-distribution; red arrowheads: NP-Sec61ß co-distribution; blue arrowheads: Sec61ß-PI4P co-distribution. For each condition, the graph on the right corresponds to a fluorescence intensity profile for NP (red), Sec61ß (green), and PI4P (cyan) along the yellow line drawn in the inset, starting from the knob. **B–D.** U2OS-Sec61ß-mEmerald cells treated as in (A) were analyzed with the Fiji software to determine the number of PI4P punctae (B), the percentage of PI4P punctae associated to the ER (C), and the percentage of PI4P punctae associated to NP puncta (D) in individual cells. Each dot represents one cell, and the data from three independent experiments are shown (black, gray, and blue dots). The mean and standard deviation values are represented as histograms (62–70 cells per condition). (B, C) ****: *p*-value < 0.0001, ns: not significant, two-way ANOVA. (D) ****: *p*-value < 0.0001, unpaired *t* test. nd: not determined, because the diffuse cytoplasmic distribution of NP signal in cells treated with the siRNA targeting RAB11A precludes analysis of the percentage of PI4P punctae associated to NP punctae. **E.** U2OS-Sec61ß-mEmerald cells were infected with WSN at a MOI of 5 PFU/cell for 8 h, or mock-infected. Fixed cells were stained for ATG16L1 and NP, and nuclei were stained with DAPI (white). Cells were imaged with a confocal microscope. Whole cell scale bar: 5 µm; insets scale bar: 1 µm. arrowheads: NP-Sec61ß-ATG16L1 co-distribution. For each condition, the graph on the right corresponds to a fluorescence intensity profile for Sec61ß (green), ATG16L1 (red), and NP (cyan) along the yellow line drawn in the inset, starting from the knob. **F.** U2OS-Sec61ß-mEmerald cells treated as in (E) were analyzed with machine learning segmentation and Fiji software to determine the percentage of ER area positive for ATG16L1. Each dot represents the mean of multiple cells from one image, and the data from five independent experiments are shown (white, black, gray, blue and light blue dots). The mean and standard deviation values are represented (17 images per condition). ****: *p*-value < 0.0001, unpaired *t* test. The data underlying this figure can be found at https://zenodo.org/records/15682874 (raw images) and S6 File (graphs raw data).

## ATG16L1 depletion or drug-mediated inhibition of PI4KIIIA activity leads to reduced viral progeny release

In ATG16L1-depleted cells infected at a high MOI (5 PFU/cell) and immunostained for the NP protein at 8 hpi, the mean intensity of the NP signal at the vicinity of the plasma membrane is reduced about 2-fold compared to control cells (Fig 6A and 6B), ($p < 0.0001$), while the mean intensity of the intracellular NP signal is increased ($p > 0.0001$) (Fig 6A–6C). The most likely interpretation for this pattern is an impaired transport of vRNPs to the plasma membrane and their subsequent accumulation in the cytoplasm, in agreement with the observed reduction in viral egress in single-cycle conditions (S7E Fig). In multicycle conditions, the production of infectious virions is reduced ~10-fold upon ATG16L1 depletion, compared to ~100-fold ($p < 0.05$) upon RAB11A depeletion (Fig 6D). Altogether, our observations indicate that ATG16L1 is involved at late stages of IAV life cycle and promotes vRNP transport to the plasma membrane, concomitant with or possibly through the control of local PI4P production on ER membranes (Fig 5).

To further confirm the importance of PI4P for the late stages of the viral cycle, we examined the impact of GSK-A1-mediated depletion of PI4P. GSK-A1, a selective inhibitor of the PI4KIIIA kinase complex which includes FAM126A [39], was added 2h post-infection with the WSN virus at a high MOI. At 6 hpi, the steady-state levels of the HA, NP, and NS1 proteins are unaffected by the presence of GSK-A1 (S9A and S9B Fig), ruling out any major toxicity or inhibition of early phases of the viral cycle. The mean intensity of the NP signal at the vicinity of the plasma membrane is reduced in GSK-A1-treated cells compared to control cells (Fig 6E and 6F) ($p < 0.001$). The trend is the same as in ATG16L1-depleted cells but the reduction is less pronounced upon GSK-A1 treatment, consistent with the fact that no significant change is measured regarding the mean intensity of the intracellular NP signal (Fig 6G). Finally, the amount of infectious viral particles produced in the supernatant is reduced ~4.5-fold ($p < 0.05$) in GSK-A1-treated cells compared to DMSO-treated cells (Fig 6H). These observations further support the functional importance of the rewiring of intracellular PI4P to ER membranes for vRNP egress.

## Discussion

The first evidence that RAB11A is essential for the cytoplasmic transport of influenza vRNPs and the production of infectious virions were provided in the early 2010s [7,23–25]. Despite progress in characterizing these late stages of the viral life cycle (e.g., [10–13]), the precise mechanisms by which RAB11A is involved remain elusive. Here, we show that IAV infection strongly modifies the proximity interactome of RAB11A, contributes to an alteration of the balance between phosphoinosides PI3P and PI4P, and induces a cross-talk between RAB11A, ATG16L1, and PI4P at the vicinity of the ER, which promotes vRNP transport.

Among the >1900 proteins whose abundance in RAB11A proximity interactome was decreased upon infection, we found the RAB11FIP-1, -2, -3, -4, and -5 proteins, i.e., RAB11A effector proteins essential to the endosomal recycling process. This finding is consistent with previous reports that the efficiency of RAB11A-dependent recycling pathway is altered in IAV-infected cells [9,10] and that depletion of RAB11FIP-2 or RAB11FIP-3 has no detectable impact on the production of infectious viral particles [24]. In addition, the abundance of 28 RAB proteins (notably RAB5A, RAB5C, RAB7A) and RAB effector proteins (for instance RABGEF1) was found to be decreased in the RAB11A proximity interactome of infected cells compared to control cells, while the opposite trend was observed for RAB2B, RAB23, and RAB43 (S2 File). Overall, our observations reinforce the notion that in IAV-infected cells, the intracellular localization, interaction with other endocytic and exocytic compartments and function of RAB11A-positive membranes are distinct from those of recycling endosomes [12].

In contrast, RAB11A proximity interactome in IAV-infected cells is enriched in proteins (FAM126A, INPP5K, ATG16L1) known to stimulate the production of PI4P, a key player of the stress-dependent regulation of endomembrane morphodynamics [35,36,40]. We show that concomitantly, IAV infection induces a significant decrease in PI3P as well as a significant increase of PI4P cellular levels, in a ATG16L1-dependent manner. Treatment with a selective inhibitor of the type III

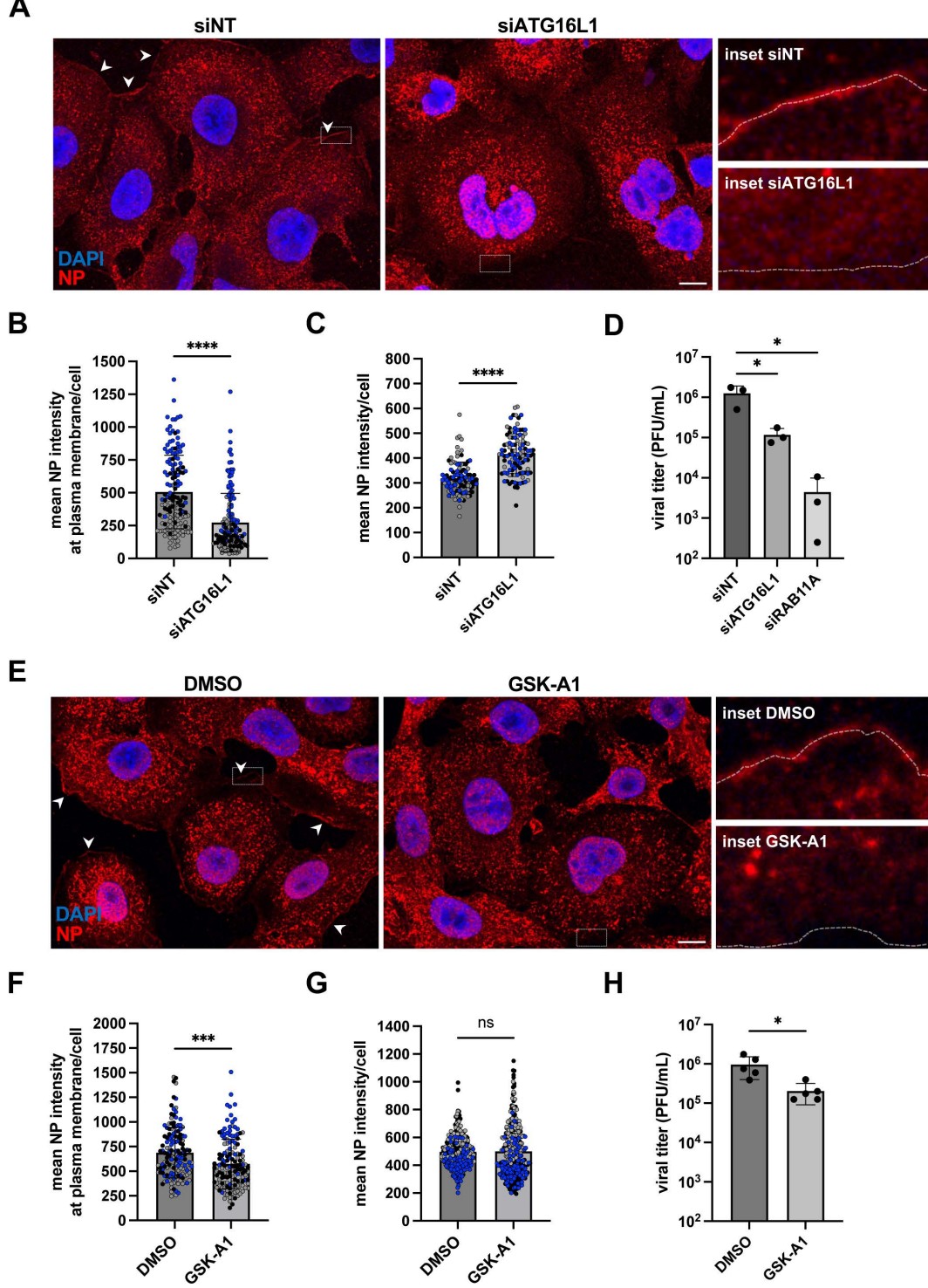

**Fig 6. Impact of ATG16L1 depletion or GSK-A1 treatment on vRNP egress and viral progeny. A.** A549 cells were treated with the indicated siRNAs for 48 h and subsequently infected with WSN at a MOI of 5 PFU/mL. At 8 hpi, fixed cells were stained for NP and nuclei were stained with DAPI (blue). Cells were imaged with a confocal microscope. Accumulations of NP signal at the plasma membrane are indicated by white arrowheads. Scale bar: 10 µm. The dotted white lines in the insets delineate the cell border. **B.** A549 cells were treated as in (A) and analyzed with the Fiji software to measure the mean NP signal along a line drawn over the plasma membrane in individual cells. Each dot represents one cell, and the data from three independent experiments are shown (black, gray, and blue dots). The median and standard deviation values are represented (149 cells per condition). ****: *p*-value

< 0.0001, ns: not significant, unpaired *t* test. **C.** A549 cells treated as in (A) were analyzed with the Fiji software to determine the mean intensity of NP signal per cell. Each dot represents one cell, and the data from three independent experiments are shown (black, gray, and blue dots). The median and standard deviation values are represented (100–106 cells per condition). ****: *p*-value < 0.0001, unpaired *t* test. **D.** A549 cells were treated with the indicated siRNAs for 48 h and subsequently infected with WSN at a MOI of 0.001 PFU/mL. At 24 hpi, the supernatants were collected and the infectious titers were determined by plaque assay. The mean ± SD of three independent experiments is shown. *: *p*-value < 0.05, paired *t* test. **E.** A549 cells were infected with WSN at a MOI of 5 PFU/mL. Two hours later, the PI4KIIIA inhibitor GSK-A1 drug was added at a final concentration of 100 nM. DMSO at the same concentration was used as a control. At 6 hpi, fixed cells were stained for NP and nuclei were stained with DAPI (blue). Cells were imaged with a confocal microscope. Accumulations of NP signal at the plasma membrane are indicated by white arrowheads. Scale bar: 10 μm. The dotted white lines in the insets delineate the cell border. **F.** A549 cells were treated as in (E) and analyzed with the Fiji software to measure the mean NP signal along a line drawn over the plasma membrane in individual cells. Each dot represents one cell, and the data from three independent experiments are shown (black, gray, and blue dots). The median and standard deviation values are represented (147 cells per conditions): ***: *p*-value < 0.001, unpaired *t* test. **G.** A549 cells treated as in (E) were analyzed with the Fiji software to determine the mean intensity of NP signal per cell. Each dot represents one cell, and the data from three independent experiments are shown (black, gray, and blue dots). The median and standard deviation values are represented (318–343 cells per condition). ns: non significant, unpaired t test. **H.** A549 cells were infected with WSN at a MOI of 5 PFU/mL. Two hours later, the GSK-A1 drug was added at a final concentration of 100 nM. DMSO at the same concentration was used as a control. At 6 hpi, the supernatants were collected and the infectious titers were determined by plaque assay. The mean ± SD of 5 independent experiments is shown. *: *p*-value < 0.05, paired *t* test. The data underlying this figure can be found at https://zenodo.org/records/15682874 (raw images) and S6 File (graphs raw data).

PI4KA kinase, which is regulated by FAM126A, results in a delayed accumulation of vRNPs at the plasma membrane and a reduction of the production of infectious viral particles, demonstrating the functional importance of PI4P for vRNP transport. Notably, the PI4P fraction associated with ER membranes represents ~60%–70% of the total PI4P pool in IAV-infected cells, compared to ~ 20%–40% in mock-infected cells. These very novel observations, taken together with the fact that RAB11A, ATG16L1, and vRNPs are localized at the vicinity of ER membranes upon infection (this study and [10–12]), point to the ER as an essential transport platform for vRNPs.

We confirm our previous observations that IAV infection induces the remodeling and tubulation of ER membranes all throughout the cell, in a RAB11A-independent manner [12]. The viral-induced depletion in PI3P, a PI enriched in the membrane of early endosomes [41], might contribute to ER remodeling. Indeed, the reduction in PI3P levels induced by nutrient deprivation, a distinct type of cellular stress, was shown to result in a loss of contacts between early endosomes and the ER and a subsequent reshaping of the ER [42]. However, unlike the conversion of tubular membranes to sheets induced by nutrient deprivation [42], we observed an extension and tubulation of sheet membranes toward the periphery of IAV-infected cells. ER stress signaling, which was shown to be induced by the expression of IAV glycoproteins, and balanced by the NS1 viral protein through its host protein shut-off activity [43,44], could possibly contribute to ER remodeling. In addition, or alternatively, the HA and neuraminidase could directly trigger ER remodeling by inducing membrane zippering, as recently shown by in situ cryo-electron tomography [13].

Based on super-resolution microscopy and statistics-based quantification, we provide evidence that vRNPs tend to accumulate at the vicinity of ER membranes in RAB11A-depleted cells. Uncovering the RAB11A-independent mechanism by which vRNPs are targeted to the ER is beyond the scope of this study. Notably, we show that ~60% of the PI4P punctae present at the ER membranes upon IAV infection co-distribute with vRNPs. Depletion of RAB11A results in a moderate but significant increase of the PI4P associated with ER membranes, suggesting the existence of a RAB11A-PI4P cross-talk coupled with PI4P turnover. There is multiple evidence that cross-talks between PIs and RAB GTPases can control membrane remodeling and trafficking [17]. In particular, PI4P was shown to recruit the RAB11A effector KIF13A and the membrane-shaping factor BLOC1 to induce the remodeling of RAB11A-positive sorting endosomes into recycling endosomal tubules [45]. KIF13A and the BLOC1 subunit 1 (BLOC1S1) are detected at the proximity of RAB11A in mock-infected cells; they are significantly less abundant (KIF13A) or undetected (BLOC1S1) at the proximity of RAB11A in IAV-infected cells (S3 File), which reinforces the notion that the environment of RAB11A membranes is strongly altered upon infection. We speculate that in the context of IAV-infected cells, a RAB11A-PI4P cross-talk at the vicinity of ER subdomains induces the recruitment of distinct RAB11A effectors and/or membrane-shaping factors, and leads to the biogenesis of

RAB11A-positive, vRNP-coated transport vesicles. This hypothesis is consistent with the observation that vRNPs and RAB11A-positive membranes concentrate in liquid condensates at the vicinity of the ER [10,11]. Contact sites between ER membranes and RAB11A-positive membranes, or possibly membrane fusion events, are likely involved and mediate the transfer of vRNPs from the vicinity of the ER membrane to the transport vesicles.

Our data provide evidence for a pivotal role of ATG16L1 in coordinating RAB11A, vRNPs, and PI4P-enriched membranes in the process of vRNP transport. Indeed, upon infection of ATG16L1-depleted cells, the cellular PI4P levels, the proportion of PI4P punctae associated with vRNPs, the ability of vRNPs to accumulate at the plasma membrane at 8 hpi, and the production of infectious viral particles, were all significantly reduced compared to control ATG16L1-expressing cells. Beyond its role in autophagy, ATG16L1 appears as an important player in the regulation of the endomembrane system in response to stress [46]. It was notably shown to be involved in the biogenesis of the primary cilium, a sensor of external chemical and mechanical stress, by targeting the INPP5E phosphatase at the cilium membrane to produce PI4P [34]. Our data indicate that in the context of IAV-infected cells, ATG16L1 controls the identity of ER membrane subdomains in terms of PI4P content and PI4P proximity with vRNPs, and cooperates with RAB11A present at the vicinity of the ER. Although ATG16L1 is a key component of the autophagic machinery, and a cooperation between ATG16L1 and RAB11A-positive membranes has been described in the context of the canonical autophagic response to starvation stress [47], there is growing evidence that ATG16L1 also plays a significant role in membrane-based processes that are not autophagic [48]. In IAV-infected cells, the autophagy process is initiated, but its full completion is prevented. The viral NS1 protein was found to repress the formation of autophagosomes in A549 and HeLa cells [49]. In A549 cells constitutively expressing GFP-LC3, autophagosomes were detected, but their fusion with lysosomes was inhibited by the viral M2 protein [50], and there is evidence that M2 abrogates the TBC1D5-RAB7 interaction to escape lysosomal degradation [51]. Interestingly, a recent study reports that silencing of canonical autophagic proteins such as ATG2A or ULK1 does not impair influenza vRNP trafficking, whereas silencing of another component of the autophagy machinery, ATG9A, does so [10]. An increase in intracellular NP at late time points was observed upon depletion of ATG9A [10] (similar to what we observed upon depletion of ATG16L1), and ATG9A was found to regulate membrane-microtubule contacts in IAV-infected cells, leading the authors to conclude that this mechanism is unrelated to canonical autophagy. Likewise, we speculate that in the context of an IAV infection, the mode of action of ATG16L1 is most likely unrelated to canonical autophagy.

Therefore, we propose that upon IAV infection, ATG16L1 acts as a stress sensor that coordinates with RAB11A to regulate the identity of endomembranes and ensure delivery of vRNP-coated membranes to the cell surface (Fig 7). The precise sequence of protein recruitment during this process as well as the protein and lipid composition of vRNP transport vesicles are fascinating biological questions that remain to be addressed. Our findings extend to IAVs the notion that viruses have evolved strategies to modulate the metabolism and localization of cellular lipids and to create new secretory pathways by reshaping ER membranes [52]. They also highlight the role of proteins of the autophagy machinery in regulating intracellular trafficking and the hijacking of this function by pathogens.

## Materials and methods

### Cells and viruses

HEK-293T (ATCC CRL-3216), A549 (kindly provided by Pr. M. Schwemmle, University of Freiburg) and U20S-Sec61ß-mEmerald cells (engineered as in [27], kindly provided by O. Schwartz, Institut Pasteur, Paris, France) were grown in complete Dulbecco's modified Eagle's medium (DMEM, Gibco) supplemented with 10% (v:v) fetal calf serum (FCS), 100 U/mL penicillin and 100 µg/mL streptomycin. Madin-Darby Canine Kidney (MDCK) cells, kindly provided by the National Reference Center for Respiratory Viruses (Institut Pasteur, Paris, France) were grown in Modified Eagle's Medium (MEM) supplemented with 5% FCS, 100 U/mL penicillin, and 100 µg/mL streptomycin. Cultured cells were routinely tested by PCR for the presence of mycoplasma, with the primers listed in the S5 File. The recombinant viruses A/WSN/33 (WSN, [53]), WSN-PB2-Strep [54]), and A/Victoria/3/75 (VIC, [55]) were produced by reverse genetics as described in [54]. The Zika

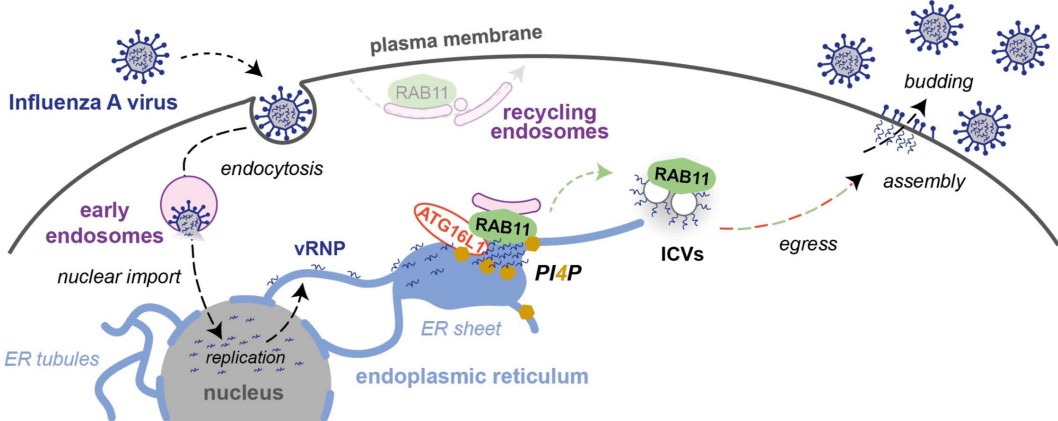

**Fig 7. Model for influenza vRNP transport from the nucleus to the plasma membrane.** Upon influenza virus entry, viral ribonucleoproteins (vRNPs) are imported into the nucleus where they serve as a template for the transcription and replication of the viral genome. The neo-synthesized vRNPs are exported from the nucleus and associate with the endoplasmic reticulum (ER) membrane. Viral infection promotes the remodeling of ER sheets and their extension toward the cell periphery. The recruitment of ATG16L1 to vRNP-enriched ER sites promotes the local production of PI4P and the recruitment of RAB11A. RAB11A, but not ATG16L1, is critical for the biogenesis of vRNP-coated vesicles previously described as irregularly coated vesicles (ICVs) [12]. In addition to RAB11A, ATG16L1 and the PI4P signaling machinery are required for the ICVs to be fully functional and efficiently transported toward the plasma membrane, where viral particles are assembled and bud from the cell surface.

strain PF13 (kindly provided by V. M. Cao-Lormeau and D. Musso, Institut Louis Malardé, Tahiti Island, French Polynesia) was isolated from a viremic patient in French Polynesia in 2013. When indicated, cells were treated with CHX (Sigma-Aldrich C4859, 100 µg/mL) from minus 1–6 hours post-infection or the GSK-A1 drug (Sigma-Aldrich SML2453, 100nM) from 2 to 6 h post-infection, or with dimethylsulfoxyde (DMSO, Sigma-Aldrich) as a control.

## Plasmids and lentiviral transduction

Reverse genetics plasmids for the WSN virus were kindly provided by G. Bronwlee (Oxford University, UK). Reverse genetics plasmids for the VIC virus [55] were developed in J. Ortin's group and kindly provided by W. Barclay (Imperial College London, UK). To generate the lentiviral pLVX-3xHA-TurboID-RAB11A plasmid, the 3xHA-TurboID coding sequence was amplified by PCR using the plasmid 3xHA-TurboID-NLS-pCDNA3 (a gift from A.Ting, Standford University, USA, Addgene #107171 [28] as a template, and subcloned instead of mCherry into pLV-CMV-mCherry-RAB11A-IRES-puro, an unpublished derivative of the pLV-CMV-eGFP lentiviral vector (Duke vector core, Duke University). To produce the corresponding lentiviruses, $10^7$ HEK-293T cells were plated in 10 cm dishes, before co-transfection the next day with 15 µg of the pLVX-3xHA-TurboID-RAB11A plasmid along with 10 and 5 µg of the pΔ8.74 (Addgene #22036) and pMD2.G (Addgene #12259) packaging plasmids, respectively, using polyethylenimine (PEI, Polysciences, 3 µL of PEI at 1 mg/mL for 1 µg of DNA). At 72 hpt, the conditioned media was passed through a 0.45µm-pore size filter and placed over A549 cells. At 48 h post-transduction, cells were split and cultured in puromycin (1 µg/mL) containing media to select for a polyclonal subpopulation of transduced cells. Cellular clones were isolated upon limiting dilution in 96-well plates. The resulting polyclonal and clonal cells were assessed for expression of 3xHA-TurboID-RAB11A by western blotting.

## Affinity purification of biotinylated proteins

The protocol for TurboID-mediated proximity labeling was an adaptation from the protocol described in [29]. Briefly, A549-TurboID-RAB11A cells were seeded in 75 cm² dishes ($10^7$ cells/dish) and infected with the WSN virus at a MOI of 5 PFU/cell or mock-infected. At 8 hpi, the culture medium was replaced with 10 mL of prewarmed (37 °C) medium supplemented

with 50 μM biotin (Sigma Aldrich, B4501). The cells were incubated at 37 °C for 10 min, washed four times with 2 mL ice-cold PBS, and dry-frozen at −80 °C for at least 1h 30 min. They were then lysed in 1.2 mL of RIPA buffer, incubated for 10 min on ice, and centrifuged for 10 min at 13,000g at 4°C. The supernatants were collected, an aliquot of 120 μL was frozen at −80°C and the remaining was incubated with 125 μL of magnetic streptavidin beads (ThermoFisher #88817, pre-washed according to the supplier's recommendations) overnight at 4 °C on a rotating wheel. The beads were washed at room temperature using a magnet, twice with 1 mL of RIPA buffer (2 min), once with 1 mL of KCl 1M (2 min), once with 1mL Na$_2$CO$_3$ (10 s), once with 1mL urea in 10 mM Tris HCl (10 s), and twice with 1mL of distilled water (2 min). Upon centrifugation, 950 μL of water was removed, and the beads covered with the remaining 50 μL were kept frozen at −80 °C until tryptic digestion on beads and LC–MS/MS analysis.

**LC–MS/MS sample preparation and data acquisition**

**Whole cell proteome.** Proteins from cell lysates were reduced using 5 mM dithiothreitol (DTT, Sigma-Aldrich #43815) for 30 min at 25°C with agitation. Following reduction, the proteins were alkylated with 20 mM iodoacetamide (IAA, Sigma-Aldrich #I114) for 30 min at 25 °C under agitation. Protein isolation and digestion were carried out using the Single-Pot Solid-Phase-enhanced Sample Preparation (SP3) method, with minor modifications from the original protocol as described by Hughes and colleagues [56]. In brief, SP3 beads were prepared by mixing hydrophilic and hydrophobic Sera-Mag SpeedBeads (GE Healthcare, Chicago, IL, USA) in a 1:1 (v/v) ratio, washed three times with water, and reconstituted to a final concentration of 50 μg/μL. To the protein samples, 8 μL of prepared beads were added, followed by the addition of anhydrous acetonitrile (ACN) to a final concentration of 75% (v/v). The samples were agitated in a thermomixer at 800 rpm for 30 min at room temperature. After 1 min on a magnet, the supernatant was removed, and the beads were washed twice with 80% ACN and once with 100% ACN. For digestion, 100 mM ammonium bicarbonate (ABC, pH 8.0) containing sequencing-grade modified trypsin (Promega, #V5111) was added to the bead-bound proteins at a protein-to-enzyme ratio of 30:1. The samples were incubated for 12 h at 37 °C. After digestion, the supernatant containing peptides was collected into a new tube following 1 min on a magnet. The beads were then washed with water, mixed for 10 min at room temperature, and the supernatant was pooled with the initial peptide solution. The resulting peptides were acidified with formic acid (FA) to a final concentration of 1%.

**Eluates enriched in biotinylated proteins.** The processing of interactomics samples was performed as per the protocol detailed by Cho and colleagues (2020) [29]. Briefly, protein attached to the beads were incubated with a digestion buffer consisting of 1 M urea, 1 mM DTT, and 50 mM Tris-HCl (pH 8.0) containing 0.4 μg trypsin, for 1 h at 25°C with agitation. The supernatants were collected into fresh tubes, and the beads were washed twice with a mild denaturing buffer containing 2 M urea and 100 mM ABC. The supernatants were pooled, reduced with 5 mM DTT for 30 min at 25°C under agitation, and alkylated with 20 mM IAA for 30 min at 25°C with agitation. The samples were diluted 2-fold with 100 mM ABC, and an additional 0.5 μg of trypsin was added for further digestion over 16 h at 37°C with agitation. The resulting peptides were acidified to 1% with FA.

**Desalting.** Digested peptides were desalted using C18 cartridges (Agilent Technologies, 5 μL bead volume, 5,190–6,532) and eluted with ACN 80%, FA 0.1%. Finally, the peptide solutions were speed-vac dried and resuspended in ACN 2%, FA 0.1% buffer. Only for global proteome samples, absorbance at 280 nm was measured with a Nanodrop 2000 spectrophotometer (Thermo Scientific) to inject an equivalent of DO = 1.

**Mass spectrometry.** A nanochromatographic system (Proxeon EASY-nLC 1,200 - Thermo Fisher Scientific) was coupled on-line to a Q Exactive Plus Mass Spectrometer (Thermo Fisher Scientific) using an integrated column oven (PRSO-V1 - Sonation GmbH, Biberach, Germany). For each sample, peptides were loaded into a capillary column picotip silica emitter tip (home-made column, 40 cm x 75 μm ID, 1.9 μm particles, 100 Å pore size, Reprosil-Pur Basic C18-HD resin, Dr. Maisch GmbH) after an equilibration step in 100% buffer A (H$_2$O, FA 0.1%). Peptides from global proteome samples were injected at constant quantity and peptides from enriched biotinylated proteins at constant volume. Peptides

were eluted with a multi-step gradient from 5% to 25% buffer B (80% ACN, FA 0.1%) during 95 min, 25% to 40% during 15 min and 40% to 95% during 10 min at a flow rate of 300 nL/min over 130 min. Column temperature was set to 60 °C.

MS data were acquired using Xcalibur software using a data-dependent Top 10 method (Global proteome) or a Top 5 method (Biotin enriched proteins) with survey scans (300–1,700 *m/z*) at a resolution of 70,000 and MS/MS scans (fixed first mass 100 *m/z*) at a resolution of 17,500. The AGC target and maximum injection time for the survey scans and the MS/MS scans were set to 3.0E6, 20 ms and 1E6, 60 ms (global proteome) or 100 ms (enriched biotinylated proteins), respectively. The isolation window was set to 1.6 *m/z* and normalized collision energy fixed to 28 for HCD fragmentation. We used a minimum AGC target of 1.0E4 for an intensity threshold of 1.7E5 (global proteome) or 1.0E5 (enriched biotinylated proteins). Unassigned precursor ion charge states as well as 1, 7, 8, and >8 charged states were rejected and peptide match was disabled. Exclude isotopes was enabled and selected ions were dynamically excluded for 45 s (global proteome) or 30 s (enriched biotinylated proteins).

**Protein identification and quantification.** MS Raw files were uploaded into MaxQuant software (version 2.0.3.0), where A549 data were searched against the Uniprot *homo sapiens* database (20,360 proteins the 13/12/2021), and WSN virus data were searched against the IAV database (10 proteins the 27/07/2022), both of which were modified to contain target sequences for TurboID. Methionine oxidation, protein N-terminal acetylation, and lysine biotinylation were variable modifications; cysteine carbamidomethylation was a fixed modification; the maximum number of modifications to a protein was 5. The minimum peptide length was set to 7 amino acids, with a maximum peptide mass of 8,000 Da. Search was performed with trypsin as specific enzyme with a maximum number of two missed cleavages. Identifications were matched between runs within replicates of a same condition. A FDR cutoff of 1% was applied at the peptide and protein levels. Peptides were quantified using unique and razor peptides. The mass spectrometry proteomics data have been deposited to the ProteomeXchange Consortium via the PRIDE partner repository [57] with the dataset identifier PXD053858.

**Statistical analysis of mass spectrometry data.** For both the global proteome and enriched biotinylated protein samples, Maxquant intensities were uploaded into our R package. To find the proteins more abundant in one condition than in another, the intensities quantified using Maxquant were compared. Reverse hits, potential contaminants, and proteins not well identified with a 1% FDR ("Only identified by site") were first removed from the analysis. Only proteins identified with at least one peptide that is not common to other proteins in the FASTA file used for the identification (at least one "unique" peptide) were kept. Additionally, only proteins quantified in at least two replicates of one of the two compared conditions were kept for further statistics. Proteins without any value in one or the other biological condition have been considered as proteins quantitatively present in a condition and absent in another. They have therefore been set aside and considered as differentially abundant proteins. They have been displayed in barplots and ranked from the one with the highest IBAQ (intensity-based absolute quantification) value to the lowest on the side of volcano plots (Figs 3A and S4B). An IBAQ value is a measure of protein abundance [58]. After this step, intensities of the remaining proteins (quantified in both conditions) were first log-transformed (log2). Next, intensity values were normalized by median centering within conditions (section 3.5 in [59]). Missing values were imputed using the impute.slsa function of the R package imp4p [60]. Statistical testing was conducted using a limma *t* test thanks to the R package limma [61]. An adaptive Benjamini-Hochberg procedure was applied on the resulting *p*-values thanks to the function adjust.p of the cp4p R package [62] using the robust method described in [63] to estimate the proportion of true null hypotheses among the set of statistical tests. These proteins were displayed in volcano plots (Figs 3A and S4B). The final set of proteins of interest is composed of the proteins which are differentially expressed according to this statistical analysis, and those which are absent from one condition and present in another (S2 and S3 Files).

Functional enrichment analyses of terms and pathways in the proteins of interest have been determined using the app stringApp [64] of Cytoscape [65]. Different backgrounds have been used for the enrichment tests: either the list of proteins identified in all the replicates of biotinylated eluates or the list of proteins identified in all the replicates of biotinylated

eluates and total lysates. A significantly low *p*-value means the proportion of proteins related to a term or pathway is significantly superior in the considered list than in the used background. The PIP-related proteins displayed in the heatmap of Fig 3B were selected because their description or GO terms mention the keyword "phosphatidylinositol".

### Immunostaining for proteins

$7.5 \times 10^4$ A549- or U2OS-derived cells were seeded on glass coverslips (13 mm diameter, #1.5, Epredia, CB00130RAC20MNZ0) in 24-well plates, and infected 24 h later with the WSN of VIC virus at a MOI of 5 PFU/cell. At the indicated time-points, cells were fixed with pre-heated (37 °C) PBS-4% PFA (Fisher Scientific, 47377) for 15 min and washed three times with 1 mL PBS. For most protein immunostainings, cells were incubated in 50 mM $NH_4Cl$ (Sigma Aldrich)-PBS for 10 min at room temperature to quench free aldehyde groups from PFA. Cells were washed three times in 1 mL PBS and incubated in a Triton X100-containing permeabilization and blocking solution (PBS supplemented with 0.1% Triton X-100, 5% Normal Donkey serum) for 1h at room temperature. Cells were then incubated overnight at 4 °C with primary antibodies directed against RAB11 (Invitrogen 71-5300, 1:100), CLIMP63 (R&D systems AF7355, 1:500), ATG16L1 (MBL PM040, 1:200), the HA tag (Invitrogen 26183, 1:100), or the influenza NP (BioRad MCA 400 or Invitrogen PA5-32242, 1:1000). Cells were then washed three times in 0,05% Tween 20-PBS and stained with an appropriate secondary antibody conjugated to an Alexa Fluor (AF) dye (Jackson Immunoresearch 715-545-150; 711-545-152 (AF488); 715-165-150 (Cy3); 711-606-152 (AF647), ThermoFisher A-21436 (AF555) 1:400) and with DAPI (ThermoFisher, 62248, 1 μg/mL) diluted in the same permeabilization and blocking solution. When indicated, DY-488-conjugated Strep-Tactin XT (IBA Lifesciences 2-1562-050, 1:200) was added in the same solution. Cells were then washed three times in 0,05% Tween 20-PBS, one time with 1 mL PBS and one time with water. Coverslips were mounted on glass slides with Fluoromount-G antifade reagent (Invitrogen, 00-4958-02).

In the case of dual stainings of ER markers CLIMP63 and RNT3 (Santa Cruz sc-374599, 1:150), together with the influenza NP or Zika NS3 antigen (anti-NS3 antibody [66] kindly provided by Andres Merits, 1:1000), cells were incubated in a saponin-containing permeabilization and blocking solution (PBS supplemented with 0.1% saponin, 1% Bovine Serum Albumin) for 30 min at room temperature. Cells were washed 3 times in 1 mL PBS for 5 min at room temperature. Free aldehyde groups from PFA were quenched using the same permeabilization and blocking solution supplemented with 20 mM glycine (Sigma Aldrich) for 15 min at room temperature. Cells were then incubated overnight at 4 °C with primary antibodies diluted in permeabilization and blocking solution. Cells were washed three times in 1 mL PBS for 5 min at room temperature prior to incubation with appropriate AF-conjugated secondary antibodies and DAPI diluted in permeabilization and blocking solution for 1 h at room temperature. Cells were washed three times in 1 mL PBS for 5 min at room temperature and coverslips were mounted on glass slides with Fluoromount-G.

In the case of samples prepared for STED microscopy, U20S Sec61ß-mEmerald cells were fixed with PBS-4% PFA (for 15 min, washed three times with 1 mL PBS, and incubated with a permeabilization and blocking solution (PBS supplemented with 0.2% Triton X-100, 0.25% fish gelatin (Sigma Aldrich, G7765) 5% Normal Donkey serum (Merck, S30) and 3% Normal Goat Serum (Merck, S26)) for 1 h at room temperature. Cells were then incubated overnight at 4 °C with primary antibodies directed against NP and GFP to amplify the Sec61ß-mEmerald signal (Abcam 13970, 1:500) in staining buffer (PBS supplemented with 0.2% Triton X-100, 0.125% fish gelatin, 5% Normal Donkey serum and 3% Normal Goat Serum). Cells were then washed three times in 0.05% Tween 20-PBS and stained with an appropriate secondary antibody conjugated to an AF dye (AF594, ThermoFisher A-11042; Atto 647N, Merck 50185) in staining buffer for 1h at room temperature. Cells were then washed three times in 0.05% Tween 20-PBS, once with 1 mL PBS and once with water. Coverslips were mounted on glass slides with Prolong Gold antifade reagent (Invitrogen, P10144).

### Immunostaining for PI4P and PI3P

Cells were fixed with pre-heated (37 °C) PBS-4% PFA (Fisher scientific, 47377) for 15 min, washed with PBS, and incubated for 20 min in blocking buffer (PBS with 5% bovine serum albumin, or PBS-5% BSA). For PI3P labeling, cells were

incubated with purified FYVE-GST recombinant protein at a final concentration or 20 μg/mL in PBS with 0,5% saponin and 5% BSA for 1 h at room temperature, washed with PBS, and incubated with an anti-GST antibody (Rockland, 600-143-200, 1:300) in PBS-5% BSA for 1 h at room temperature. For PI4P labeling, cells were incubated overnight at 4 °C with a PI4P biosensor (SNAP-SidC, [38]) or an anti-PI4P antibody (Echelon-Z-P004, 1:150) in PBS with 0,5% saponin and 5% BSA, washed with PBS, incubated for 1 h at room temperature with an appropriate secondary antibody conjugated to an AF dye (donkey anti-mouse IgG, Life Technologies), and post fixed 10 min with PBS-2% PFA.

## Confocal microscopy

For confocal microscopy, Leica TCS SP8 or Nikon Confocal AX scanning confocal microscopes equipped with HC PL APO CS2 40× (NA = 1.3) or 63× (NA = 1.4) oil objectives were used. A MAX z-projection was applied to 5–10 stacks (0.35 μm step-size) in all figures. The fluorescence signals were acquired with the LAS X software (Leica) and analyzed with Fiji (ImageJ) to determine fluorescence intensity profiles on a single focal plane. Cell Profiler [21,22] was used to determine the ratios of CLIMP63+ to RTN3+ areas. A SUM z-projection was applied to stacks of 15 images (0.35 μm step-size) and segmentation of the cells was performed by selecting the "Identify Primary Objects" module, using an adaptive, 2-class Otsu thresholding method. The number of PI4P punctae and their localization were manually analyzed using the "Cell counter" plugin in Fiji software. Line scans were recorded using the "Color Profiler" plugin (Color_Profiler. jar, https://imagej.nih.gov/ij/plugins/color-profiler.html) in Fiji software. The percentage of ER area positive for ATG16L1 was determined by applying machine learning segmentation (Ilastik software) to the ER and ATG16L1 signals, as in Da Graça and colleagues [67]. Segmented images were merged and analyzed using the "color threshold" module of the Fiji software to quantify the percentage of ER area overlapping with the ATG16L1-positive area. To determine the mean intensity of the NP signal at the plasma membrane, a segmented line was drawn along the whole plasma membrane of each cell present in a given region of interest, and the fluorescence intensity along the line was measured using the Fiji software.

## STED microscopy

STED images were acquired using the confocal laser scanning microscope LEICA SP8 STED 3DX equipped with a 93× (NA = 1.3) glycerol immersion objective and with three hybrid detectors (HyDs). The specimens were excited with a pulsed white-light laser (598 nm or 640 nm) and depleted with a pulsed 775 nm depletion laser to acquire nanoscale imaging using SMD HyD detector. STED Images (2048 × 2048 px) were averaged 16 times in line and acquired with a magnification zoom >2 leading to a pixel size in the range of 20–30 nm. Excitation laser was adjusted to avoid any saturating pixels, and the same laser intensity and HyD sensitivity were used for both control and infected cells.

The levels of association of NP with Sec61ß were analyzed using a dedicated automatized program designed by L. Danglot using the Icy software [68] and "protocol" plugin. Pearson coefficient analysis was performed using the Co-localization Studio Icy Block [69] that takes into account both diffuse and/or aggregated signals. Cell area was automatically delineated using Easy Cell shape plugin (10.5281/zenodo.4317782) that rely on HK Means segmentation. Cell area was then converted to the Region of interest (Cell ROI) and Pearson coefficient was calculated with Sec61ß and NP channels within the Cell ROI from 18 control cells and 19 infected cells (sampled from 3 independent experiments). To analyze the Pearson coefficient in the surroundings of the nuclear membrane, we delineated a "perinuclear region" band of 100 pixels around the nucleus, contained within the Cell ROI.

## Immunoblots

Total cell lysates prepared in RIPA or Laemmli buffer were loaded on 4–12% gradient acrylamide gels (Invitrogen, NP0323). Immunoblot PVDF membranes (GE Healthcare Life science, Amersham Hybond 10600023) were incubated with HRP-conjugated streptavidin (Cell Signaling technology, 3999S, 1:2000) or with primary antibodies directed against RAB11 (Invitrogen 71-5300, 1:500), CLIMP63 (R&D systems AF7355, 1:500), RTN3 (Santa Cruz sc-374599, 1:500), ATG16L1

(MBL PM040, 1:1000), PB2 (GeneTex GTX125925, 1:5,000), α-tubulin (Sigma-Aldrich, T5168 – B-5-1-2, 1:10,000), Histone-3 (Cell Signaling Technology, 9715S, 1:10,000) or A/Puerto Rico/8/34 whole virions (produced in-house) and revealed with appropriate secondary antibodies (Sigma Aldrich, A9044 and A9169, 1:10,000) and the ECL 2 substrate (Pierce). The chemiluminescence signals were acquired using the Chemidoc imaging system (Biorad) and a semi-quantitative analysis was performed with the ImageLab software (BioRad). Uncropped immunoblots are shown in S4 File.

### siRNA-based assays

Small interfering RNAs (siRNAs) (Dharmacon ON-TARGETplus SMARTpools and Non-targeting Control pool) were purchased from Horizon Discovery, or from QIAGEN in the case of ATG16L1 siRNAs (Qiagen, SI04317418 and SI04999134). To assess the production of infectious viral particles, A549 cells seeded in 96-well plates ($1.2 \times 10^4$ cells/well) were transfected with 30 nM of siRNA, using 0.3 µL of the DharmaFECT1 transfection reagent (Horizon Discovery), and infected at 48 h post-transfection (hpt) with the WSN virus at a MOI of 0.001 PFU/cell. Plaque assays were performed on MDCK cells as described in [70]. To assess subcellular features, A549 cells seeded in 24-well plates ($7.5 \times 10^4$ cells/well) were transfected with 30 nM of siRNA, using 3 µL of DharmaFECT1. At 24 hpt, cells were trypsinized, seeded on glass coverslips (13 mm diameter, #1.5, Epredia, CB00130RAC20MNZ0) in 24-well plates ($10^5$ cells/well), and infected 24 h later with the WSN virus at a MOI of 5 PFU/cell. Immunostainings were performed as described above. Cell viability in the presence of siRNAs was assessed in the 96-well plate format, using the CellTiter-Glo Luminescent Viability Assay kit (Promega). Knockdown efficiency of individual transcripts was quantified by RT-qPCR using SYBR green (ThermoFisher, 4309155) with the LightCycler 480 system (Roche), and the primers listed in S5 File, or by western blot using antibodies directed against RAB11A or ATG16L1 as indicated above. RNA levels were normalized to GAPDH and analyzed using the $2^{-\Delta\Delta CT}$ method [71].

### Viral replication assays

Cells were seeded in 96-well plates ($3 \times 10^4$ cells/well), in triplicates 24 h prior to infection. To assess the accumulation of vRNA species from infected cells, cells were infected with the WSN virus at a MOI of 5 PFU/cell. Total RNA was isolated from pooled triplicates 5 hpi with RNeasy Mini columns according to the manufacturer's instructions (RNeasy Kits, Qiagen) and strand-specific qPCRs were performed as described in [72]. Briefly, total RNA was reverse transcribed using primers specific for NP-mRNA and -vRNA or cellular glyceraldehyde 3-phosphate deshydrogenase (GADPH) with SuperScript III Reverse Transcriptase (Invitrogen), and quantified using SYBR-Green (Roche) with the LightCycler 480 system (Roche). RNA levels were normalized to GAPDH and analyzed using the $2^{-DDCT}$ method [71]. To assess the production of infectious viral particles, cells were infected with the WSN virus and incubated in OptiMEM supplemented with 0.5 µg/mL TPCK and 1% BSA (MOI = 0.001 PFU/cell) or in DMEM supplemented with 10 FCS (MOI = 5 PFU/cell). The supernatants from triplicates were pooled and titrated by a plaque assay on MDCK cells.

## Statistical analysis of cell-based data

Statistical tests were performed using the GraphPad Prism (v10) software.

For immunostaining data, unpaired Student $t$ test or two-way ANOVA were performed.

For viral PFU data, two-way ANOVA was performed and Dunnett's test was used for multiple comparisons with respect to the siNT reference.

## Supporting information

**S1 Fig. Viral-induced remodeling of the ER (related to Fig 1). A.** A549 cells were infected with WSN at a MOI of 5 PFU/cell for 8 h. Fixed cells were stained for the viral HA and cellular CLIMP63 proteins. Nuclei were stained with DAPI (blue), and cells were imaged with a confocal microscope. Scale bar: 10 µm. **B.** Fluorescence intensity profile for HA

(cyan) and CLIMP63 (magenta) along the yellow line drawn in panel (A) (merge inset), starting from the knob. **C.** A549 cells were infected with ZIKV PF13 at a MOI of 5 PFU/cell for 24 h, or mock-infected. Fixed cells were stained for the viral NS3 and the cellular RTN3 and CLIMP63 proteins. The RTN3 staining was used to delineate the cell edges. Nuclei were stained with DAPI (blue), and cells were imaged with a confocal microscope. White arrows indicate viral factories surrounded with remodeled ER membranes. Scale bar: 10 μm. The data underlying this figure can be found at https://zenodo.org/records/15682874 (raw images) and S6 File (graphs raw data).
(TIF)

**S2 Fig. Impact of RAB11A depletion on ER remodeling and vRNP localization upon infection (related to Fig 2). A.** A549 cells were infected with the WSN-PB2-Strep virus at a MOI 5 PFU/cell for 8 h. Fixed cells were stained for NP, PB2 (StrepTactin-488) and RAB11. Nuclei were stained with DAPI (blue), and cells were imaged with a confocal microscope. Scale bar: 10 μm. **B.** Fluorescence intensity profile for NP (red), PB2-Strep-tag (cyan) and RAB11 (magenta) along the yellow line drawn in panel (A) (merge inset), starting from the knob. **C.** A549 cells were treated with RAB11A-specific or control Non-Target (NT) siRNAs for 48 h, and subsequently infected with WSN at a MOI of 5 PFU/cell for 8 h, or mock infected. Fixed cells were stained for the viral NP and cellular RTN3 and CLIMP63 proteins. The difference in permeabilisation protocols (saponin versus Triton) most likely accounts for the difference in NP signal patterns in this experiment compared to the experiment shown in Fig 2A. Scale bar: 10 μm. The data underlying this figure can be found at https://zenodo.org/records/15682874 (raw images) and S6 File (graphs raw data).
(TIF)

**S3 Fig. RAB11A proximity labeling (related to Fig 3). A–C.** Characterization of the clonal population of A549 cells stably expressing TurboID-RAB11A. (A) Total cell lysates of A549 parental cells (untransduced), a polyclonal population of transduced A549- TurboID-RAB11A cells, and the clonal population isolated thererof, were analyzed by western blot using an antibody specific for RAB11. The two panels derive from one and the same membrane that was hybridized with an antibody specific for RAB11, thereby allowing to visualize both the endogenous RAB11 and overexpressed TurboID-RAB11A proteins. (B) The clonal A549-TurboID-RAB11A cells were infected with the WSN virus at a MOI of 5 FPU/cell for 8 h. Fixed cells were stained for the HA-tag (3xHA-TurboID-RAB11A, magenta) and the viral NP (red). Nuclei were stained with DAPI (blue), and cells were imaged with an epifluorescence microscope. Scale bar: 10 μm. (C) Total cell lysates of parental A549 cells or clonal A549-TurboID-RAB11A cells, incubated or not for 10 mn at 37 °C in the presence of 50 μM biotin, were analyzed by western blot using streptavidin conjugated with HRP to detect biotinylated proteins. **D.** Correlation matrices between replicates of eluates (left) and total lysates (right). A correlation matrix represents the Pearson correlation coefficients between each pair of samples computed using all complete pairs of intensity values measured in these samples. Intensity values correspond to TMT-MS2 quantitative relative abundance metrics in the columns titled "Reporter intensity corrected" of the "proteinGroups.txt" file of MaxQuant. The samples identification numbers are indicated in the format "Mock.number of the technical replicate" and "IAV.number of the technical replicate". Pearson correlation coefficients are indicated in the lower triangular parts of the matrices. In the upper triangular parts, thediameters and gradient colors of the circles are function of these coefficients. **E.** Distributions of the log2(intensities) for the proteins without missing values in the eluates (left) and total lysates (right). The samples identification numbers are indicated on the vertical axis in the format "[Mock].number of the technical replicate" and "[Virus].number of the technical replicate". The data underlying this figure can be found at https://zenodo.org/records/15682874 (raw images), in S3 File (mass spectrometry data analyses), S4 File (uncropped western blots) and S6 File (graphs raw data).
(TIF)

**S4 Fig. RAB11A proximity labeling (related to Fig 3). A.** Gene Ontology (GO) term enrichment analysis of the set of 3,775 biotinylated proteins identified as being enriched in the biotynylated samples across all conditions and replicates, over a total of 6011 proteins identified in biotinylated eluates and/or total lysates across all conditions and replicates (both

sets of proteins are listed in the S1 File). The graph represents the number of genes corresponding to each indicated category (x axis) and the enrichment p-value (color scale). GO terms related to membranes and to intracellular transport/localization/vesicles have been highlighted using blue and green color fonts, respectively. **B.** Volcano plot showing the log2 fold change (x axis) and its significance (−log10(p-value), y axis) associated to a False Discovery Rate <1%) for each protein (dots) in total lysates from the RAB11A proximity labeling experiment. The log2 fold change refers to the enrichment in WSN-infected ($n = 4$) versus mock-infected ($n = 4$) samples. Blue and green dots represent proteins enriched in WSN-infected versus mock-infected samples, and proteins enriched in mock-infected versus WSN-infected samples, respectively. The iBAQ plots shown on the sides of the volcano plot provide additional information on proteins for which no statistical comparison of the abundance could be performed (hence they are not represented in the volcano plot), because they are present only in WSN-infected samples (blue) or only in mock-infected samples (green). The data underlying this figure can be found in S1 File (GO term enrichment) and S3 File (volcano plot).
(TIF)

**S5 Fig. Viral-induced perturbations of PI4P levels monitored by immuno-fluorescence (related to Fig 4). A.** A549 cells were infected with WSN at a MOI of 5 PFU/cell for 8 h, or mock-infected. Fixed cells were stained for PI4P with a specific antibody. Nuclei were stained with DAPI (blue), and cells were imaged with a confocal microscope. Scale bar: 10 µm. **B.** A549 cells treated as in (A) were analyzed with the Fiji software to determine PI4P mean intensity per cell. Each dot represents one cell, and the data from three independent experiments are shown (black, gray and blue dots). The median and standard deviation values are represented (154–170 cells per condition). ****: p-value < 0.0001, unpaired t test. **C.** U2OS-Sec61ß-mEmerald cells were infected with WSN at a MOI of 5 PFU/cell for 8 h, or mock-infected. Fixed cells were stained for PI4P and nuclei were stained with DAPI (white). Cells were imaged with a confocal microscope. Scale bar: 5 µm. **D.** U2OS-Sec61ß-mEmerald cells treated as in (C) were analyzed with the Fiji software to determine the percentage of the total PI4P punctae associated to ER in individual cells. Each dot represents one cell, and the data from three independent experiments are shown (black, gray and blue dots). The mean and standard deviation values are represented as histograms (63–66 cells per condition). ****: p-value < 0.0001, unpaired t test. **E and G.** A549 cells were treated with control non-target (NT) siRNAs or with siRNAs targeting FAM126A, INPP5K (E) or ATG16L1 (G) for 48 h, and subsequently infected with WSN at a MOI of 5 PFU/cell for 8 h, or mock-infected. Fixed cells were stained for PI4P with a specific antibody. Nuclei were stained with DAPI (blue), and cells were imaged with a confocal microscope. Scale bar: 10 µm. **F and H.** A549 cells treated as in E and G, respectively, were analyzed with the Fiji software to determine the mean intensity of the PI4P signal per cell. Each dot represents one cell, and the data from three independent experiments are shown (black, gray and blue dots). The median and standard deviation values are represented (101–145 cells per condition). ***: p-value < 0.001, ****: p-value < 0.0001, ns: non significant, unpaired t test. The data underlying this figure can be found at https://zenodo.org/records/15682874 (raw images) and S6 File (graphs raw data).
(TIF)

**S6 Fig. siRNA-mediated knock-down efficiency and cell viability assessment. A**. A549 cells were treated with RAB11A-specific or control Non-Target (NT) siRNAs for 48 h, and subsequently infected with WSN at a MOI of 5 PFU/cell for 8 h or mock-infected. Fixed cells were stained for the viral NP and the cellular RAB11 proteins. Nuclei were stained with DAPI (blue), and cells were imaged with a confocal microscope. Scale bar: 10 µm. **B, C.** Knock-down efficiency of siRNA pools. A549 cells were treated with the indicated siRNAs for 48 hours. (B) Total cell lysates were prepared and analyzed by western blot, using the indicated antibodies. (C) Total RNA were extracted and analyzed by RTqPCR using gene specific primers. The residual mRNA levels are expressed as percentages (100%: NT siRNA). Data shown are the mean ± SD of three experiments performed in triplicates. ****: p-value < 0.0001 (one-way ANOVA and Dunnett's multiple comparisons test, reference: NT siRNA). **D.** Cell viability upon treatment with siRNA pools. A549 cells were treated with the indicated siRNAs for 48 h and cell viability was determined at 48 hpt using the CellTiter-Glo Luminescent Viability

Assay (Promega). The data shown (RLU: Relative Light Units) are expressed as percentages (100%: NT siRNA) and are the mean ± SD of three independent experiments performed in triplicates. Black and white dots correspond to two distinct series of experiments, in which the "Death Control siRNA" (Qiagen) and a siRNA directed against PLK1 were used as positive controls, respectively. The dotted line indicates a 20% reduction in luciferase signal. ****: $p < 0.0001$ (one-way ANOVA and Dunnett's multiple comparisons test, reference: NT siRNA, no indication means no significant difference). The data underlying this figure can be found at https://zenodo.org/records/15682874 (raw images), S4 File (uncropped western blots) and S6 File (graphs raw data).
(TIF)

**S7 Fig. Impact of ATG16L1, FAM126A and INPP5K depletion on early, intermediate and late stages of IAV infection (related to** Fig 4**). A, B.** A549 cells were treated with the indicated siRNA for 48 h and subsequently infected with WSN at a MOI of 5 PFU/cell, in the presence (+) or absence (−) of cycloheximide (CHX). (A) At 6 hpi, total cell lysates were prepared and the steady-state levels of viral proteins were analyzed by western blot, to control for the efficiency of CHX treatment (no viral protein expression). (B) At 6 hpi, total RNA were extracted and the levels of NP-mRNAs and NP-vRNAs were analyzed by strand-specific RTqPCR. The results of three independent experiments, labeled #1, #2, and #3 in (A), are shown. Dotted line: background for mRNA detection in mock-infected cells. Two-way ANOVA with Dunnett's multiple comparison test; using the siNT without/with CHX samples as a reference for the other without/with CHX samples, respectively, revealed no statistically significant differences. **C, D.** A549 cells were treated with the indicated siRNA for 48 h and subsequently infected with WSN at a MOI of 5 PFU/cell or mock-infected. At 5 hpi, total cell lysates were prepared and the steady-state levels of viral proteins were analyzed by western blot. (C) Cropped blots of three independent experiments, labeled #1, #2, and #3, are shown. (D) The signals for HA, NP ad NS1 were normalized over the α-tubulin signal (α-Tub) and expressed as percentages (100%: NT siRNA). The data shown are the mean ± SD of the three independent experiments. *: $p$-value < 0.05, **: $p$-value < 0.01, ***: $p$-value < 0.001 (two-way ANOVA with Dunnett's multiple comparison test, reference: siNT). **E.** A549 cells were treated with the indicated siRNA for 48 h and subsequently infected with WSN at a MOI of 5 PFU/mL. At 3 or 4 hpi, the culture medium was extensively washed away four times to remove the viral input and replaced with fresh medium. One hour later, i.e., at 4 or 5 hpi, respectively, the supernatants were collected. The last wash (residual input) and collected supernatants were titrated by a plaque assay. The titers were normalized over the control condition (NT siRNA). The data shown are the mean ± SD of the three independent experiments. *: $p$-value < 0.05, **: $p$-value < 0.01, ***: $p$-value < 0.001 (two-way ANOVA with Dunnett's multiple comparison test, reference: siNT). The dotted lines represent the average ratio value for residual inputs. **F.** A549 cells were infected with WSN at a MOI of 5 PFU/cell for 4 h, or mock-infected. Fixed cells were stained for PI4P using the SNAP-SidC probe. Nuclei were stained with DAPI (blue), and cells were imaged with a confocal microscope. Scale bar: 10 µm. **G.** A549 cells treated as in (F) were analyzed with the Fiji software to determine the mean intensity of the PI4P signal per cell. Each dot represents one cell, and the data from three independent experiments are shown (black, gray and blue dots). The median and standard deviation values are represented (115–144 cells per condition). ****: $p$-value < 0.0001, unpaired $t$ test. The data underlying this figure can be found at https://zenodo.org/records/15682874 (raw images), S4 File (uncropped western blots) and S6 File (graphs raw data).
(TIF)

**S8 Fig. Impact of ATG16L1 depletion on the distribution of RAB11A and vRNPs (related to** Fig 5**). A.** A549 cells were treated with control NT or ATG16L1 siRNA for 48 h and subsequently infected with WSN at a MOI of 5 PFU/cell for 8 h. Fixed cells were stained for viral NP and cellular RAB11 proteins. Nuclei were stained with DAPI (blue), and cells were imaged with a confocal microscope. Scale bar: 10 µm. **B.** Fluorescence intensity profile for NP (red) and RAB11 (cyan) along the white line drawn in panel (A) (merge inset), starting from the knob. The data underlying this figure can be found at https://zenodo.org/records/15682874 (raw images) and S6 File (graphs raw data).
(TIF)

**S9 Fig. Impact of GSK-A1 treatment on the steady-state levels of viral proteins (related to** Fig 6**). A.** A549 cells were infected with WSN at a MOI of 5 PFU/cell. Two hours later, the GSK-A1 drug was added at a final concentration of 100 nM. At 6 hpi, total cell lysates from five independent experiments (#1 to #5) were prepared and analyzed by western blot, using the indicated antibodies. Cropped blots are shown. **B.** The signals for the viral HA, NP and NS1 proteins and the RTN3 protein were normalized over the histone H3 (HH3) signal and expressed as percentages (100%: mock-infected cells). The data shown are the mean ± SD of three independent experiments (two-way ANOVA with Dunnett's multiple comparison test, reference: mock-infected cells, no indication means no significant difference). The data underlying this figure can be found in S4 File (uncropped western blots) and S6 File (graphs raw data).
(TIF)

**S1 File. Proteins detected by mass spectrometry in the lysates and eluates from the proximity labeling experiment.** Separate datasets are listed in separate tabs, as defined in the first tab labeled "S1_File".
(XLSX)

**S2 File. Differential analysis of mass spectrometry data from eluates of the proximity labeling experiment.** Separate datasets are listed in separate tabs, as defined in the first tab labeled "S2_File".
(XLSX)

**S3 File. Differential analysis of mass spectrometry data from lysates of the proximity labeling experiment.** Separate datasets are listed in separate tabs, as defined in the first tab labeled "S3_File".
(XLSX)

**S4 File. Uncropped western blot membranes.** Set of membranes corresponding to distinct figure panels showing western blot analysis are listed in separate tabs, which are labeled with the corresponding figure panel number.
(XLSX)

**S5 File. Primers sequences.** The sequence of primers used for RTqPCR assessment of knock-down efficency and for testing for mycoplasma presence are indicated in two separate tabs.
(XLSX)

**S6 File. Raw data corresponding to the graphs.** Datasets corresponding to distinct Figure panels showing a graph are listed in separate tabs, which are labeled with the corresponding figure panel number.
(XLSX)

## Acknowledgments

We thank Federica Ferrentino and Ulrike Eggert (King's College London, United Kingdom) for their very helpful advice regarding Cell Profiler-based analyses, and Maxime Chazal and Nolwenn Jouvenet (Institut Pasteur, Paris, France) for infection experiments with the Zika virus. We thank Olivier Schwartz, Blandine Monel and Julian Buchrieser (Institut Pasteur, Paris, France) for providing U2OS cells stably expressing Sec61ß-mEmerald, Andres Merits (University of Tartu, Estonia) for providing the antibody against Zika virus NS3, Wendy Barclay (Imperial College London, United Kingdom) and Caroline Goujon (IRIM, Montpellier, France) for providing the A/Victoria/3/75 reverse genetics plasmids.

We gratefully acknowledge the technical and scientific support from the UtechS Photonic Bioimaging (Imagopole, C2RT, Institut Pasteur, supported by ANR-1N-INBS-04; Investments for the Future), the Necker SFR technical platforms and especially Meriem Garfa-Traoré at the Cell imaging facility, and the NeurImag Imaging core Facility team (part of IPNP, Inserm U1266 and Université Paris Cité, and member of the national infrastructure France-BioImaging supported by the French National Research Agency, ANR-10-INBS-04). We thank the Leducq foundation for supporting the acquisition of the Leica SP8 Confocal/STED 3DX microscope.

## Author contributions

**Conceptualization:** Lydia Danglot, Etienne Morel, Jean-Baptiste Brault, Nadia Naffakh.

**Formal analysis:** Carla Alemany, Juliane Da Graça, Quentin Giai Gianetto, Maud Dupont, Lydia Danglot.

**Funding acquisition:** Lydia Danglot, Mariette Matondo, Etienne Morel, Nadia Naffakh.

**Investigation:** Carla Alemany, Juliane Da Graça, Maud Dupont, Sylvain Paisant, Thibaut Douché, Catherine Isel, Lydia Danglot, Jean-Baptiste Brault.

**Project administration:** Nadia Naffakh.

**Resources:** Cédric Delevoye.

**Supervision:** Cédric Delevoye, Mariette Matondo, Etienne Morel, Jean-Baptiste Brault, Nadia Naffakh.

**Validation:** Carla Alemany, Juliane Da Graça, Quentin Giai Gianetto, Lydia Danglot, Mariette Matondo, Etienne Morel, Jean-Baptiste Brault, Nadia Naffakh.

**Visualization:** Carla Alemany, Juliane Da Graça, Quentin Giai Gianetto, Maud Dupont, Thibaut Douché, Lydia Danglot, Etienne Morel, Jean-Baptiste Brault, Nadia Naffakh.

**Writing – original draft:** Carla Alemany, Juliane Da Graça, Quentin Giai Gianetto, Thibaut Douché, Lydia Danglot, Mariette Matondo, Etienne Morel, Jean-Baptiste Brault, Nadia Naffakh.

**Writing – review & editing:** Carla Alemany, Juliane Da Graça, Quentin Giai Gianetto, Thibaut Douché, Catherine Isel, Cédric Delevoye, Lydia Danglot, Mariette Matondo, Etienne Morel, Jean-Baptiste Brault, Nadia Naffakh.

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
