## [Editor Report · Decision Letter 0]

Dear Dr Naffakh, 

Thank you for submitting your manuscript entitled "Influenza A virus-induced production of PI4P at the endoplasmic reticulum involves ATG16L1 and promotes the egress of viral ribonucleoproteins." for consideration as a Research Article by PLOS Biology.

Your manuscript has now been evaluated by the PLOS Biology editorial staff, as well as by an academic editor with relevant expertise, and I am writing to let you know that we would like to send your submission out for external peer review.

Once your full submission is complete, your paper will undergo a series of checks in preparation for peer review. After your manuscript has passed the checks it will be sent out for review. To provide the metadata for your submission, please Login to Editorial Manager (https://www.editorialmanager.com/pbiology) within two working days, i.e. by Nov 28 2024 11:59PM.

Kind regards,

Melissa

Melissa Vazquez Hernandez, Ph.D.

Associate Editor

PLOS Biology

---

## [Decision Letter · Decision Letter 1]

Dear Dr Naffakh,

Thank you for your patience while your manuscript "Influenza A virus-induced production of PI4P at the endoplasmic reticulum involves ATG16L1 and promotes the egress of viral ribonucleoproteins." was peer-reviewed at PLOS Biology. It has now been evaluated by the PLOS Biology editors, an Academic Editor with relevant expertise, and by several independent reviewers. 

In light of the reviews, which you will find at the end of this email, we would like to invite you to revise the work to thoroughly address the reviewers' reports. While the reviewers appreciated the study's relevance and novelty, they raised specific concerns requiring attention. Reviewer 1 noted that the functional relevance and underlying mechanisms were not fully explored, particularly the impact of RAB11A KD on the punctate staining of viral NP, alternative explanations for reduced viral progeny, and increased PI4P levels. Reviewer 2 indicated that the conclusions based on Figure 4 are not well-supported and suggested confirming PI4P levels using an alternative method. Reviewer 3 identified issues with the data in Figure 6 and recommended repeating experiments with a higher MOI to evaluate late stages of viral replication; they also stated that the data in Figure 5E does not adequately support the conclusion regarding ATG16L localization changes. After consulting the Academic Editor, we concur with these concerns and request additional experimental revisions, particularly to eliminate alternative explanations for Figures 4 and 6, confirm PI4P levels, and use higher MOI to assess late replication stages. Addressing these points will significantly strengthen your work.

Given the extent of revision needed, we cannot make a decision about publication until we have seen the revised manuscript and your response to the reviewers' comments. Your revised manuscript is likely to be sent for further evaluation by all or a subset of the reviewers.

**IMPORTANT - SUBMITTING YOUR REVISION**

*Re-submission Checklist*

*Published Peer Review*

*PLOS Data Policy*

*Blot and Gel Data Policy*

Sincerely,

Melissa

Melissa Vazquez Hernandez, Ph.D.

Associate Editor

PLOS Biology

REVIEWERS' COMMENTS:

Reviewer #1: 

This manuscript is well written and contains high quality data. It includes a number of correlative data showing that influenza virus infection leads to changes in phosphoinositide metabolism and recruitment to ER domains. The effects on PI metabolism seem to require the activities of PI4P processing enzymes as well as RAB11A and ATG16L1. Many of the observations made by the authors are interesting, however their functional relevance or underlying mechanisms have not been fully explored. I mainly suggest rewording and revisiting some of the conclusions drawn by the authors.

RAB11A KD , but not ATG16L1 KD, is shown to abolish the punctate staining of viral NP (Fig.'s 2A and 5A). This observation has not been fully explored. Perhaps the authors could at least comment on this finding which seems to stand out in their data.

Figures 4&6: Could some of the effects on reduced viral progeny release resulting from the absence of ATG16L1 or PI4P associated enzymes be due to effects on viral entry or degradation? Similarly, would it be possible that, in the presence of siRNA targeting FAM126A, NPP5K and ATG16L1, the suppressed increase in PI4P levels is due to changes in viral uptake? These possibilities have not been explored or commented on in this manuscript.

Additional comments:

The authors say that "comparable steady-state levels for the recombinant TurboID-RAB11A and the endogenous RAB11A proteins, as assessed by western-blot (S3A Fig)." not clear this statement can be derived from the shown blots unless the authors can present an uncropped blot containing both endo-RAB11A and exogenous protein.

The conclusion "this proviral activity of ATG16L1 is most likely mediated by the control of local PI4P production on ER membranes." should be toned down as the authors show no direct evidence for this and the conclusion is based on correlative data.

Fig 6C; there seems to be an increase of NP-associated puncta; is this due to reduce viral exit/degradation?

The conclusion that influenza virus trafficking is perturbed in the absence of ATG16L1 or PI4P processing should be supported by colocalisation experiments with various endocytic compartments. Alternatively, such conclusions can be toned down.

The authors conclude on pg.17 that "mode of action of ATG16L1 could be unrelated to canonical autophagy" without giving any experimental evidence. They suggest that the "RAB11A interactome" did not detect any canonical autophagy players, such as ATG5-12 or WIPI2 . This is not a valid reasoning as the RAB11A proximity labelling depends on the location of the tag and distance to neighboring proteins and is not necessarily a reflection of the interactome. This conclusion should be revisited.

Reviewer #2: 

The study by Alemany et al. aims to shed light on the still poorly understood process of influenza A virus (IAV) vRNP transport to assembly sites. By performing proximity labelling of RAB11A during infection the authors reveal that IAV can modulate levels of the phosphoinositide PI4P. They show that this modulation of PI4P levels is required for vRNP transport. They also reveal a role for ATG16L1 in regulating PI4P dynamics and coordinating RAB11A-dependent vRNP transport. 

The data are presented in a very clear manner and support the conclusions drawn, with one exception (see points on figure 4). In my opinion, this work represents an important step forward in our understanding of the cell biology of IAV infection. 

Major comments:

- Fig. 4A-B: Immunofluorescence-based detection is useful for analyzing co-localization but in my opinion, it is not ideal for quantification of protein or PI4P levels given its limited dynamic range. Can the authors confirm the increase in PI4P levels by a different method? This would strengthen this crucial finding of the manuscript substantially. 

- Fig. 4: The images seem to show much larger differences between mock and infected cells than the quantifications. Furthermore, it seems that in the quantifications the difference is driven by the replicate with the blue dots. Can the authors confirm that they observed a difference in mock versus infected cells when analyzing each experiment separately? Maybe the authors should also consider showing cells with a less prominent effect that is more reflective of the quantification results. Given that the results shown in fig. 4 are central for the conclusions this part should be strengthened. 

Minor comments:

- Fig. S4A: The GO terms are difficult to read, and I would appreciate some more annotation. For example, the authors could mark the GO terms that refer to membrane-associated proteins, given that they say these are overrepresented. 

- Fig. 3A, S4B: In my opinion, the green and the blue plots next to the volcano plot do not provide or illustrate any information, other than giving the number of proteins found in that category. Either it needs to be explained what the panels add, or they should be deleted. 

Reviewer #3: 

This manuscript from Alemany and Da Graça shows a novel role for ATG16L1 and P14P in the influenza infection. This work uses high resolution microscopy to further establish the perturbation of the endoplasmic reticulum upon influenza infection, and a proximity-ligation based proteomics approach to characterize the Rab11A interactome in the presence/absence of IAV. They show that IAV infection results in a change in the relative PI3P/P14P levels, and the P14P punctae colocalize with NP at late stages of infection in a ATG16L1 dependent manner. The work is thoughtfully designed, the writing is clear and the microscopy is striking. The findings will be of broad interest to both influenza virologists and cellular biologists and contains several important observations. The characterization of the Rab11A interactome is thorough (four replicates greatly increase the utility of this dataset) and will be valuable to the field. In addition, the finding that vRNPs make their way to the ER in a Rab11A independent manner is novel and presents a new challenge to understanding the cellular factors involved in IAV trafficking. I have identified a few issues with quantification and MOI, addressing these would strengthen the claims made in the manuscript.

Major Points

* The data in Figure 6 does not unambiguously show that the effect of ATG16L1 knockdown is occurring at a late stage of viral infection. By performing infections at a very low MOI, multiple cycles of replication are occurring (and the defect in PFU production could theoretically occur at any stage of the life cycle). While the IF data attempts to solve this problem by looking for NP accumulation, not every cell displays a uniform accumulation of NP along the entire cell periphery. The quantification here is hard to fully understand, since cells are classified in a binary fashion (by eye?), and only the percent of cells with any NP at the PM is shown in the graph (ie, only three numbers). My comparison of cells in which I think there is NP at the plasma membrane does not fully agree with the white stars used to denote 'cells with PM NP' in Fig 6E/F so the authors should probably reanalyze this in a way that does not depend on a 'by eye' determination. But really, in order to make the claim that this defect in infectious virus production occurs at a late stage in viral infection, the authors should perform synchronized infections at a high MOI, harvest virus after a single cycle of infection, and show that protein production is unaffected by western blot (as in S5D, to rule out a defect in any of the stages prior to assembly/trafficking).

* I agree, the data shows that the loss of ATG16L1 reduces the number of P14P punctua per cell (based on the quantification, the representative image shown in 5A though looks pretty similar for all three conditions in the P14P channel). However the image shown in 5E does not convincingly make the point that ATG16L changes its localization upon infection- the authors should perform the same rigorous analysis of ATG16L ER-associated punctae as was done in 5C. 

Minor Points

* The data with ZIKV distracts from the main story (both ER sheets and ER tubules appear to be disrupted, as opposed to just ER sheets for IAV). I would consider removing it.

* Figure 1B/2D, the legend mentions black, white and gray dots, but I only see black, gray and blue dots (double check this in all figures that use this schematic).

* Figure 2A, I am confused why non-specific Rab11 nuclear staining is enhanced upon Rab11A depletion. It seems plausible to me that this could be bleed through from the NP antibody, and it would help to show the staining for uninfected, depleted cells to demonstrate that the non-specific nuclear staining occurs to the same level in the absence of any NP. The depletion looks convincing by western blot, but this IF image is not as strong an argument for efficient Rab11A depletion.

---

## [Editor Report · Decision Letter 2]

Dear Nadia,

It was nice meeting you in Paris! Thank you for your patience while we considered your revised manuscript "Influenza A virus-induced production of PI4P at the endoplasmic reticulum involves ATG16L1 and promotes the egress of viral ribonucleoproteins." for publication as a Research Article at PLOS Biology. This revised version of your manuscript has been evaluated by the PLOS Biology editors and the Academic Editor.

Based on our Academic Editor's assessment of your revision, we are likely to accept this manuscript for publication, provided you satisfactorily address the remaining editorial points raised by the reviewers. Please also make sure to address the following data and other policy-related requests.

a) We routinely suggest changes to titles to ensure maximum accessibility for a broad, non-specialist readership, and to ensure they reflect the contents of the paper. In this case, we would suggest a minor edit to the title, as follows. Please ensure you change both the manuscript file and the online submission system, as they need to match for final acceptance:

"Influenza A virus induces PI4P production at the endoplasmic reticulum in an ATG16L1-dependent manner to promote the egress of viral ribonucleoproteins"

Please supply the numerical values either in the a supplementary file or as a permanent DOI’d deposition for the following figures:

Figure 1BD, 2BD, 3AB, 4BDFH, 5A-E, 6BCDEGH, S1B, S2B, S3DE, S4AB, S5BDFH, S6BD, S7BDEG, S8B, S9B

c) Please cite the location of the data clearly in all relevant main and supplementary Figure legends, e.g. “The data underlying this Figure can be found in S1 Data” or “The data underlying this Figure can be found in https://doi.org/10.5281/zenodo.XXXXX”

d) As some of your findings rely on the microscopy images, we would like to encourage you to submit all other microscopy pictures to Zenodo or FigShare so they can be available to the readers too for figures 1A, 2AC, 3ACEG, 5AE, 6AE, S1AC, S2AC, S3B, S5ACEG, S6A, S7F, S8A

e) Please ensure that your Data Statement in the submission system accurately describes where your data can be found and is in final format, as it will be published as written there.

f) Per journal policy, if you have generated any custom code during the course of this investigation, please make it available without restrictions upon publication. Please ensure that the code is sufficiently well documented and reusable, and that your Data Statement in the Editorial Manager submission system accurately describes where your code can be found.

We expect to receive your revised manuscript within two weeks. 

*Published Peer Review History*

*Press*

Sincerely,

Melissa

Melissa Vazquez Hernandez, Ph.D.

Associate Editor

PLOS Biology

---

## [Editor Report · Decision Letter 3]

Dear Nadia,

Apologies for the delay. Thank you for the submission of your revised Research Article "Influenza A virus induces PI4P production at the endoplasmic reticulum in an ATG16L1-dependent manner to promote the egress of viral ribonucleoproteins" for publication in PLOS Biology. On behalf of my colleagues and the Academic Editor, Andrew Mehle, I am pleased to say that we can in principle accept your manuscript for publication, provided you address any remaining formatting and reporting issues. These will be detailed in an email you should receive within 2-3 business days from our colleagues in the journal operations team; no action is required from you until then. Please note that we will not be able to formally accept your manuscript and schedule it for publication until you have completed any requested changes.

PRESS

Sincerely, 

Melissa

Melissa Vazquez Hernandez, Ph.D., Ph.D.

Associate Editor

PLOS Biology
